# The Use of Ozone Technology: An Eco–Friendly Method for the Sanitization of the Dairy Supply Chain

**DOI:** 10.3390/foods12050987

**Published:** 2023-02-26

**Authors:** Rinaldo Botondi, Micaela Lembo, Cristian Carboni, Vanessa Eramo

**Affiliations:** 1Department for Innovation in Biological, Agro-Food and Forest Systems, University of Tuscia, 01100 Viterbo, Italy; 2Industrie De Nora Spa, 20134 Milano, Italy

**Keywords:** ozone, dairy supply chain, cheese, product quality, microbial control

## Abstract

The dairy field has considerable economic relevance in the agri-food system, but also has the need to develop new ‘green’ supply chain actions to ensure that sustainable products are in line with consumer requirements. In recent years, the dairy farming industry has generally improved in terms of equipment and product performance, but innovation must be linked to traditional product specifications. During cheese ripening, the storage areas and the direct contact of the cheese with the wood must be carefully managed because the proliferation of contaminating microorganisms, parasites, and insects increases significantly and product quality quickly declines, notably from a sensory level. The use of ozone (as gas or as ozonated water) can be effective for sanitizing air, water, and surfaces in contact with food, and its use can also be extended to the treatment of waste and process water. Ozone is easily generated and is eco-sustainable as it tends to disappear in a short time, leaving no residues of ozone. However, its oxidation potential can lead to the peroxidation of cheese polyunsaturated fatty acids. In this review we intend to investigate the use of ozone in the dairy sector, selecting the studies that have been most relevant over the last years.

## 1. Introduction

### Overview

The environmental impact from the production of food is a focus of interest at the global level due to its relationship with climate change and competition with existing natural and energy resources [1].

Cheese is a product of great importance in the dairy sector, notably in the western States, and for this reason, today, it has a globalized origin, ranging from Europe to the USA [2]. The COVID-19 pandemic has led to a global increase in food consumption, including cheese. The global cheese market was valued at USD 72.26 billion in 2020, with a production volume of 21.69 million metric tons compared to 20.98 million metric tons in 2019. The global cheese market is expected to grow from $192.81 billion in 2021 to $211.02 billion in 2022 at a compound annual growth rate (CAGR) of 9.4%. The market is expected to grow to $289.57 billion by 2026 at a compound annual growth rate (CAGR) of 8.2% [3]. The world’s largest cheese producer in 2021 is Europe (EU-27) with 10 million tons, followed by 6 million tons in the USA [4] (Table 1).

Dairies in Germany accounted for the highest share of EU production of all main fresh and manufactured dairy products, including drinking milk (19.3% of the EU total), butter (21.0%), cheese (22.9%), and acidified milk products (23.7%) [5]. In 2021, Germany was the largest producer of cheese globally, with 2,461,334 tons. France was the second largest producer of cheese, recording a value of 1,716,120 tons, while Italy was the third largest producer of cheese globally, recording a value of 1,197,390 [6] (Table 2).

According to the 2022 Industry Sheet drawn up by Ismea, the dairy chain has strong economic importance in the national agri-food system. However, weaknesses in the supply chain has resulted in insufficient diffusion and implementation of measures to combat climate change that can be transformed into opportunities, such as the creation of sustainable products that respond to consumer needs [7]. In the dairy sector, several problems involve huge costs, economic losses, and excessive product waste. For the disinfection of the premises and equipment, a large amount of water and steam at high temperatures and chlorine-based chemicals is used. For example, ultra-high-temperature (UHT)-processed milk uses a lot of chemicals (alkaline cleaners, such as NaOH, and HNO_3_) for cleaning [8]. This results in further problems and costs for the disposal of wastewater, both for the sanitizers used, and for the production residues. Dairy wastewater (DWW) contains high levels of mineral and organic compounds, which can collect in soil and water, affecting serious environmental pollution [9]. The huge amounts of wastewater with high organic content must be disposed of and, conventionally, they are purified by physical–chemical and biological methods. Some authors reported a generation of 1–10 L of wastewater per liter of processed milk [10,11].

About 192.5 × 106 m^3^ of dairy wastewater (DWW) are generated per year in EU-27 countries, 49% of which is to produce cheese, while 19%, 18%, and 13% are required to produce drinking milk, acidified milk, and butterfat products, respectively. Germany, France, Italy, Poland, Spain, and the Netherlands produced more than 73% of DWW [12].

The treatment of wastewater with ozone allows for the production of effluents at a level that can be discharged into natural aquatic systems [13,14]. The common practice for the sanitation of water is the use of chlorine gas (Cl_2_), chlorine dioxide (ClO_2_), or hypochlorite (ClO^−^) due to the bactericidal and oxidative properties of chlorine. Ozone is a viable alternative for sanitization concerning other traditional sanitizers, such as chlorine, for over-chemical residues and potential environmental impacts that consumers care about [15].

The amount of chlorine required for sanitization applications depends on the concentration in water of the species to be oxidized, organic substances, micro-organisms, ferrous ions and manganese, nitrites, and hydrogen sulfide [16]. The legislative framework regulating the quality of water for human consumption has recently been updated with EU directive 2020/2184 of 21 December 2021 [17], which replaces the 98/83 directive [18]. The new directive provides for the monitoring of some by-products of chlorine disinfection, such as chlorates and haloacetic acids five (HAA5-monochloroacetic acid, dichloroacetic acid, trichloroacetic acid, monobromoacetic acid, and dibromoacetic acid).

The cleaning of equipment was once completely manual in the dairy sector, using brushes and detergent solutions, and it involved dismantling the equipment to access every surface. This approach was labor- and time-intensive and often quite counterproductive, since contamination can be reintroduced to the system by imperfectly cleaned equipment. These issues were solved by the introduction of CIP (Cleaning In Place) programs, which adopted automated cleaning systems in various parts of the processing plant to achieve the necessary cleaning and sanitation results [19]. However, in sanitation systems, it is advisable to try to lower the environmental impact of the various operations.

Milk and dairy products offer highly nutritious food for microorganisms to multiply, so they can produce spoilage. The microbial contamination depends on the raw material quality, the conditions under which the products were produced, and the temperature and duration of their storage [20,21], resulting in product decay due to contamination by spoilage microorganisms, bringing the product to be unsaleable or greatly depreciated. In the case of pathogenic microorganisms, the product is not marketable, as it is harmful to human health. The control of contamination by spoilage bacteria, like *Pseudomonas* spp., and pathogenic bacteria, such as *Listeria monocytogenes*, and *Salmonella* spp., is a big challenge for the dairy industry [22].

The many varieties of cheese found are characterized by different flavors, textures, and shapes, and are produced using milk, rennet, salt, and lactic ferments. The microbiota, made up of lactic acid bacteria (LAB) [23], contributes significantly to the quality and safety of the final product, even to the nutritional composition [24], developing desirable characteristics and flavor compounds, and it inhibits the growth of spoilage and pathogenic bacteria in cheese [25]. Moreover, their variation leads to obtaining a different quality of the product and, for this reason, some studies were conducted to isolate microbial strains that were obtained from the best natural fermentations [26]. Examples of starter LAB dominant during the fermentation period are *Lactococcus* and *Streptococcus* [25], while during the maturation process the dominant species are *Lactobacillus helveticus, Lactobacillus paracasei* subsp. *paracasei, Lactobacillus delbrueckii* subsp. *lactis, Enterobacter faecium*, and *Lactobacillus plantarum*, qualitatively appreciated by consumers thanks to flavor intensity and texture, as well as overall acceptability [27,28,29]. Since the inoculation of starter cultures has been shown by several studies to have a better impact on the overall quality of the cheese, there are commercial starter cultures with which to inoculate cheeses [26].

The transport of bulk milk from the dairy to the processing plant is critical, as well as the various stages of storage and processing, because it could significantly alter the milk microbiota due to contamination or the growth of microflora [30,31,32,33,34,35].

Product losses also occur due to infestation problems, such as mites, which infest the ripening rooms through the wooden boards used to mature the cheeses, with consequences ranging from product weight loss to the waste of the entire cheese. The mites are very small, hardly visible to the naked eye, and are practically ubiquitous in the dairy industry and, more generally, in the food sectors in which a product maturation phase is foreseen [36]. They are present in the wooden shelves used in the maturing rooms and spread through the clothes of the operators, or they can be transported by air; dirty environments favor their proliferation [36,37,38]. Therefore, cleaning and disinfection are key operations for food safety, but they produce a high environmental impact due to water use and wastewater. In the care areas, chemicals such as chlorine, quaternary ammonium compounds, etc., are used. Health and environmental concerns with chemical use on food products or food contact facilities are supporting the need for alternative sanitation technologies. In this sense, the potential utility of ozone lies in the fact that ozone is a stronger oxidant compound than chlorine, and it has been shown to be effective over a wider spectrum of microorganisms. Furthermore, ozone reacts with some organic compounds in food matrices, and the possible by-products are aldehydes, ketones, or carboxylic acids, which do not pose a threat to human health [39].

Unlike other disinfectants, it leaves no ozone residue and degrades to molecular oxygen upon reaction or natural degradation [40] (Figure 1).

Ozonation is a clean technique with antimicrobial power due to its oxidation potential able to reduce the microorganisms, limiting the production of enzymes, but the effectiveness of ozone can be affected by the temperature, pH, humidity, and the amount of organic matter around the cells [41].

The aim of this review is to give an overview of the current state of knowledge on several manufacturing and storage processes of the dairy chain, highlighting the potential and limiting the use of gaseous or ozonated water against the pathogenic and spoilage microflora, parasites, and insects, such as molds and mites. Specifically, we intend to start from the raw material of the dairy supply chain, the milk, up to the different final products, including their quality and sensory parameters, passing through the sanitization of the working surface areas and equipment and through the treatment of wastewater. The process flow diagram of the dairy industry is shown below (Figure 2).

## 2. Ozone (O_3_)

### 2.1. Generalities, Properties, and Legislation

Ozone (from the Greek ozein, which has an odor) is a molecule made up of three oxygen atoms arranged in space with a trigonal planar geometry with a bond angle of 116.8° and a bond length of 1.278 Å. Ozone has a molecular weight of 48 g mol^−1^ at 1 atm of pressure, a boiling point of −111.9 °C (beyond which it appears as a dark blue liquid), and a melting point of −192.7 °C. At room temperature and normal pressure, ozone is present in the gaseous state and is characterized by a pungent odor (the fresh and clean smell, typical of the air after a storm, is due precisely to the ozone that has just been generated in the atmosphere) and bluish in color if generated from dry air [37].

The inactivation mechanism against microorganisms through ozone can be obtained in two ways, oxidizing amino acids and sulfhydryl groups of peptides, proteins, and enzymes to produce small peptides at the time of ozone exposure, or oxidizing the poly-unsaturated fatty acids, with the formation of acid peroxides [42,43].

Ozone is soluble in aqueous solution (from 0 to 30 °C its solubility is thirteen times higher than that of oxygen molecule), in a logic that is inversely proportional to the temperature (solubility decreasing as the temperature of the water increases) [37].

In industrial practice, ozone generators are of three different types: corona generator, electrochemical, and UV radiation. The most used is the ozone generator through the corona effect as compared to the other methods. Three methods are used for the determination of ozone: physical, based on the measurement of some ozone properties, such as absorbance in the visible or infrared range; physical–chemical: it measures the effect of the reaction of ozone with certain reagents including chemiluminescence and heat reaction; chemical: it measures the number of reaction products between ozone and a specific reagent (an example is the indigo colorimetric method) or the reduction of the molecular weight of a polymer the ozone adds to the double bonds causing the discoloration, which is then measured with the spectrophotometer as a change in absorbance [44].

Connected to the antimicrobial power of ozone molecule is the oxidation potential of well −2.07 V; higher than that of chlorine (−1.36 V) and hydrogen peroxide (−1.78 V), and lower than fluorine (which has the highest redox potential with −2.87 V), hydroxyl radical, persulphate ion, and atomic oxygen. In nature it is present in the ozonosphere, part of the stratosphere, and is concentrated at about 25 km from sea level, and its concentration in the atmosphere is about 0.04 ppm (1 ppm = 1.96 mg m^−3^), which remains constant thanks to the dynamic balance existing between the formation and photolysis processes. The formation is due to UV radiation with a wavelength of 242 nm which dissociates molecular oxygen into atomic oxygen [37].

Ozone is not a radical species, but it is a very reactive molecule: in water it decomposes very quickly, through mechanisms giving rise to a series of reactive ROS (Reactive Oxygen Species), including the superoxide radical (O_2_•^−^), hydroxyl radical (OH•), and hydrogen peroxide (H_2_O_2_), which, in turn, react with the biological macromolecules, causing modifications of the structure and therefore of the functions. Its reactivity depends on the electronic configuration of the molecule: one of the terminal oxygen atoms is characterized by a lack of electrons and therefore by an electrophilic behavior, the other by a negative charge, which gives the molecule a nucleophilic character. It is an unstable molecule characterized by a very short half-life, greater in the gaseous state than in the aqueous solution; its decomposition is so rapid in the aqueous phase of food that its antimicrobial action takes place mainly on the surface [44]. The stability of the molecule is influenced, among other things, by the purity of the water: it has been seen observed at 20 °C that 50% of the ozone decomposes in twenty minutes if in the presence of distilled or tap water, a percentage which is reduced to 10% if double-distilled water is used. At pH 7, as the temperature increases from 15 °C to 30 °C, the half-life decreases from 30 to 12 min; moreover, the half-life in water is much shorter than in air. Taking, for example, the temperature of 20 °C, the half-life is three days in the air and only twenty minutes in water at pH 7 [45].

Having a strong pro-oxidant effect, ozone in contact with some materials could lose effectiveness, but also generate corrosion. The compatibility of this molecule obviously depends, in a directly proportional manner, on its concentration; if this remains as dry gas in a range of 2% by weight, it is compatible with most plastic materials [46] used in the food industry: PTFE, PVDF, PVC, ECTFE–Polytetrafluoroethylene, Polyvinylidene fluoride, Polyvinyl chloride, and Ethylene Chlorotrifluoroethylene. 316L and 304L stainless steel proved to be more resistant to ozone than to chlorine [47]; natural rubber is instead very susceptible, while silicone tolerates short periods of exposure well, but if exposed for long periods it oxidizes.

Since 2001, the FDA (Food and Drug Administration) has recognized the use of ozone as an antimicrobial agent in the gaseous phase or in aqueous solution in food production processes, defining this as GRAS (Generally Recognized As Safe) element, i.e., a food additive considered safe for human health [48].

Disinfectants such as ozone can be manufactured, placed on the market, and used in the European Union exclusively on the basis of an authorization issued under Regulation 528/2012 [49] and/or under the national legislation currently in force in each Member State. There are also directives on some specific uses of ozone, such as the directive 2003/40/EC [50] that defines the conditions for using ozone-enriched air for separating compounds of iron, manganese, sulfur, and arsenic from natural mineral waters or spring waters, and the labeling requirements for waters which have undergone such treatment.

The legislation related to ozone must also stop dealing with issues related to safety in the workplace as the effects related to oxidation make this molecule potentially toxic, in a logic that remains dependent on the concentration and exposure time. The human olfactory perceptibility threshold for ozone is very low (0.02–0.04 ppm in air); for exposures to concentrations of 0.1 ppm in air, irritation to the eyes, nose, and throat can be generated; one-hour exposures to increasing concentrations of 2, 4, and 15 ppm, respectively, lead to the appearance of symptoms, irritant, and finally toxic effects, while higher concentrations can have lethal effects on humans [44]. The Occupational Safety and Health Administration (OSHA) has approved, as the concentration limit for ozone gas, 0.1 ppm with an exposure time of 8 h at the workplace in the food industry [51]. Advice on the treatment of air with ozone in cheese-maturing environments from the National Committee for Food Safety of the Italian Ministry of Health (CNSA) states that operators must not be exposed to more than 0.1 ppm ozone in 8 h or more than 0.3 ppm twice a day for 15 min [37]. However, ozone treatments with high concentrations can be undertaken with no operators or during the weekly closing days [52,53].

### 2.2. Factors Affecting the Industrial Utilization of Ozone

In the food industry, chlorine and hydrogen peroxide-based sanitizers are the most used. Although their antimicrobial action is extremely effective, Khadre et al. [54] have shown that they release residues that can have a potentially toxic effect; hence the need to look for alternatives that can be just as effective, but without leading to the formation of residues. Therefore, the introduction of ozone as a sanitizer in the food sector must be seen in this context. Being characterized by a strong redox potential, ozone has excellent capacity as a disinfectant, but, having a short half-life, it decomposes rapidly into non-toxic products and therefore leaves no residues that could have an impact on the environment, much less on human health. This is also clearly correlated to a reduction in costs for the company, who will not have to deal with the disposal of residues of chemical substances used for sanitization. Furthermore, being generated on-site and on-demand, both the environmental and economic impact is further reduced, in fact, there is no need for transport and much less storage (as otherwise happens for conventional products). Finally, the costs to be incurred for the operation of ozonation systems are also low, as they require relatively low levels of electricity to operate [55].

To prevent microbial contamination in several industrial fields, disinfectants are used, such as quaternary ammonium salts, iodine, acids, and chlorine-based compounds, compounds which, although particularly effective in disinfection, release toxic substances into the environment. UV radiation can also be used (to which problems of release of radioactive substances can be associated, which accounts for more marginal use in the food sector) in addition to heat treatment [56]. The use of hot water or steam is a very effective system, but excessive heat can damage the equipment, and the associated energy cost is by no means negligible. For these reasons, ozone presents itself as a valid alternative.

Today, ozone is currently used in water for food sanitization, but also for swimming pools, showers, irrigation systems, and water purification; it is used in the environment in the gaseous phase, mainly for the elimination of volatile molecules (connected to the development of odors) and toxins [57].

Natural Organic Matters (NOM) are micro-organic pollutants that can affect the efficacy of drinking water treatment processes and the safety of drinking water. Advanced removal of water NOM by PEB (Pre-ozonation, Enhanced coagulation, and Bio-augmented Granular Activated Carbon) was studied. The results showed that the pre-ozonation treatment with 0.5 mg of O_3_ per mg of TOC (Total Organic Carbon) resulted in a TOC removal of 87%, with a concentration of 0.8 mg L^−1^ in the finished water, lower than the standard limit for chlorination treatment (2 mg L^−1^) with no formation of bromate as a carcinogenic byproduct [58].

In poultry farming, it is used to sanitize the environment from pathogenic microorganisms, to eliminate ammonia fumes and for deodorization, as well as to improve the oxygenation of the environment, reduce the risk of cross-contamination, and improve the hygienic–sanitary conditions of the feed. Ozone is also used in the preservation of fish and meat to reduce the microbial load in storage cold rooms, and it avoids contamination of products with bad smells; however, it is advisable to keep the concentration of ozone under control, as it can be a vehicle for an increase in lipid peroxidation, which causes the deterioration of the final product [59]. Additionally, for storing cereals and grains in silos, it is used as a fumigant to inhibit the formation of molds (with consequent mitigation of the risk of mycotoxin release), but also for insects that can infest stored cereals [60].

A final significant use is its use in fruit and vegetable production, including applications for the disinfection of the water used during the sanitization of products, both in the storage and conservation phases, by acting as an antimicrobial inside the storage rooms [61,62,63,64].

Additionally, due to the recent Coronavirus epidemic (COVID-19), whose transmission occurs through droplets, more attention is being given to fruit and vegetables that are bought for household consumption.

Therefore, it has become even more important to sanitize these products immediately after their harvest. Ozone could be used in this phase, as it appears to have a strong virucidal activity and is capable of inactivating viruses and bacteria [42].

However, an important factor to consider is the ability of ozone to penetrate food, which can affect the quality of the product. Therefore, a careful balance must be maintained between food safety and the maintenance of nutritional aspects [43].

In this sense, the results of the work of Shynkaryk et al. [65] have demonstrated that such changes due to ozone are limited to the product surface only, without penetration into the observed mass of material.

An interesting study on the penetration capacity of ozone for the disinfection of textile materials shows that, for effective decontamination, it is important to use ozone concentrations to allow for reaching hard-to-reach regions of different types of garments, considering the distance between the garments used [66].

### 2.3. Mechanism of Action of Ozone on Different Matrices

The way ozone acts against microorganisms is a complex process. Ozone can attack various cell membrane constituents-cell walls, the cytoplasm, endospore coats, virus capsids, and viral envelopes [54,67]. The double bonds of unsaturated fatty acids are particularly susceptible to ozone [56]. The effective antimicrobial properties of the molecule are its high oxidation potential and its ability to spread through biological membranes [68].

Due to its high oxidation potential and its ability to diffuse across biological membranes, ozone oxidizes the cellular components of the bacterial cell wall and, once it enters the cells, it oxidizes all essential components: enzymes, proteins, deoxyribonucleic acid (DNA), and ribonucleic acid (RNA). The result of this process is the destruction of the cell. The mechanism of action is, therefore, different from that of other disinfectants: chlorine, for example, penetrates the cell by diffusion and, once inside, it affects the various enzymes [45].

Ozone acts in two different ways: a direct way, in which the ozone molecule itself reacts with the organic or inorganic substance and an indirect way, in which the radicals, mainly the hydroxyl radical, are obtained following the decomposition of ozone to react with organic matter. In direct reactions, ozone acts as a dipole, with electrophilic and nucleophilic properties: its action is very selective. In the case of addition reaction to the double bond (mostly in reference to unsaturated fatty acids), a compound called ozonoid is formed, which in turn degenerates in proton solution (for example in water) into aldehyde, ketone, or zwitterion; the latter degenerates into hydrogen peroxide and carboxyl residues. As far as indirect reactions are concerned, three phases can be identified: activation, propagation, and termination. In the first phase, in the presence of an activator, such as the hydroperoxide radical (HO_2_^•^), the decomposition of ozone accelerates: at a pH higher than the value of 4.8 (corresponding to the pKa of this radical) the radical forms the anion superoxide, which triggers the chain propagation by reacting with the ozone to form the hydroxyl radical, which in turn reacts with the ozone, determining the formation of the HO_2_ radical, which will trigger the process again. In general, indirect oxidation can be applied to many organic pollutants, while direct oxidation can be applied to many inorganic pollutants, and it is no coincidence that its first and main uses concern water treatment [45,69]. Ozone acts at different levels and the mechanisms of inactivation of bacterial cells are many. As far as the cell walls and membranes are concerned, it oxidizes various components and, in particular, the unsaturated fatty acids, the enzymes present at the membrane level, the glycoproteins, and glycolipids, leading to the modification of the permeability of the cell membrane and finally to the cell lysis. The bacterial spores are protected from the action of the ozone thanks to the bark which itself, in the first instance, undergoes the oxidizing action; cellular enzymes are inactivated by O_3_ in a very effective and less selective way compared to other disinfectants, such as chlorine [59].

Other conditions being equal, the efficiency of ozone treatment varies depending on the different susceptibility of the target microorganisms. There are numerous studies that demonstrate its efficiency against both Gram-negative, less resistant, and Gram-positive bacteria, as well as against viruses, yeasts and molds, protozoam and parasites, including mites [37].

Finch et al. [70] determined concentration and exposure time to obtain the inactivation of *E. coli*; Morandi et al. [71] positively evaluated its efficiency for the control of *Listeria monocytogenes* on some cheeses. Brodowska et al. [72] showed that vegetative cells are much more susceptible to treatment than sporogens. In the study, *Bacillus cereus*, *Bacillus subtilis*, *Bacillus pumilus*, *Escherichia coli*, *Pseudomonas fluorescens*, *Eupenicillium cinnamopurpureum*, and *Aspergillus niger* were examined, and the results showed that longer treatments were required precisely for the inactivation of the microorganisms belonging to the genus *Bacillus*. Moore et al. [73] reported the efficacy of ozone against various bacteria of food interest, underlining that in the presence of biofilm this is limited; Farooq et al. [74] focused on the study of the inactivation of *Candida parapsilosis*. Recently, ozone has also been shown to be effective against *Aspergillus fumigatus*, a difficult fungus to inactivate using other methods [75]. The researchers demonstrated the effectiveness of ozone in completely removing the microbiota on the tested swatches (20 ppm for 4 min) in a test ozone chamber.

In summary, it turns out that molds are more resistant than yeasts which, in turn, are more resistant than viruses and bacteria and, among these, the most susceptible are Gram-negatives. Moreover, both bacterial and fungal spores are more resistant to the antibacterial action of ozone. Efficacy is also interesting in detoxification: mycotoxins are completely degraded or modified so that their bioactivity is compromised. Moreover, it has been seen that treatment with ozone can degrade biofilms [55]. It has been seen that the inactivation mechanisms of viruses, although so far less studied, differ from those of bacteria: viruses, in fact, do not undergo cellular lysis, but the inactivation of specific viral receptors which allow them to form cells gami with the wall of the cell to be invaded, thus allowing the virus to enter it. The result is the arrest of the viral reproduction mechanism already at the level of cellular invasion. The spectrum of action, depending, among other things, on the exposure time and applied concentration, is quite broad [37].

## 3. Ozone in the Dairy Industry

### 3.1. Microbiological: A Critical Control Point to Be Monitored

Some phases within the dairy industry are particularly critical from a microbiological point of view. Examples in the production of cheese are the thermization or pasteurization of milk which must be sufficient to inactivate all human pathogenic bacteria, as well as reduce the endogenous microbial load [76]; the syneresis which may be more or less marked, influencing the a_w_ (water activity index) and therefore the possibility of development of the various microorganisms; the salting phase which determines a change in the microbiological habitat and, therefore, also the variation of the microbial ecosystem (from the population of starter lactic acid bacteria, SLAB, to that of non-starter bacteria, NSLAB); stewing, during which the optimal environment is recreated between temperature and relative humidity for the proliferation of the microorganisms responsible for the acidification of the “pasta”; finally, the maturation phase is a fundamental one in which the NSLAB microorganisms will be responsible for the changes that will occur in the cheese and which will influence the consistency, the sensory properties of the product, and its shelf-life. For the maturation of some aged cheeses, a further criticality must be considered: the presence of mites, arthropods that tend to settle in the wooden boards on which the cheeses are placed to age, on which the parasites feed, causing serious economic losses to the companies [77].

A fundamental element for the control of microbial loads is the disinfection of systems, instruments, and rooms: the microorganisms that can colonize environments are in fact many and companies carry out cleaning operations every day using water vapor at high temperature and pressure and chemical products that are chlorine-based; these are costly operations from both an economic and an environmental point of view. Furthermore, wastewater is rich in organic matter, an element that makes it difficult and expensive to dispose of, as well as representing an excellent growth medium for microflora.

The genus *Pseudomonas* includes spoilage bacteria, responsible for undesirable odors and flavors or unusual pigments of foods [78], often isolated at several stages of the dairy production environment (*P. fluorescens*, *P. koreensis*, *P. marginalis*, *P. rhodesiae*, *P.fragi*, *P. putida*, *P. entomophila*, *P. mendocina*, and *P. aeruginosa*) [79].

The genus *Listeria* comprises *L. monocytogenes*, a Gram-positive foodborne pathogen that can form biofilms and persist for a long period on surfaces and food processing environments, being able to cause frequent contaminations of the finished products [80,81,82].

The genus *Salmonella* comprises pathogenic bacteria found in various types of foods, like fresh and fermented dairy products, adapting to an acid environment [83,84,85,86].

### 3.2. Ozone Technology in Dairy Farming

Ozone is used upstream of the dairy supply chain, starting from animal husbandry: it is an alternative to traditional methods, which involve the use of hot water and chemical products, for the sanitization of the pipes that carry the milk from the rooms of milking to collection tanks. Using ozone reduces costs (in reference to the purchase of disinfectants and the use of hot water) and the environmental impact (as it does not release residues); moreover, the effectiveness of ozonized water in the disinfection of animals, surfaces and tools has been seen. In livestock farms, the ozone can be used for the sanitization of the air in the stables: at low concentrations, it contributes to the elimination of bad odors and any pathogens present in the air [26]. Ogata and Nagahata [87] studied its potential in the treatment of bovine mastitis, a problem that afflicts farms, causing important economic losses: according to the researchers, as many as 60% of treated mastitis heals without the use of antibiotics (6–30 mg of ozone).

### 3.3. Efficacy of Ozone on the Sanitization of the Working Surface Areas and Equipment

Hot water with chemicals is usually used for cleaning and disinfection processes, generating large energy and chemicals consumptions. Many studies evaluated the use of ozonated water and gas to clean and sanitize equipment and different surfaces in dairy locations [55], (Table 3).

Guzel-Seydim et al. [88] evaluated the use of ozone for the treatment of stainless-steel surfaces to remove milk residues, demonstrating its greater effectiveness compared to the traditional treatment with hot water at 40 °C. 15 min treatments using either warm water (40 °C) or ozonated cold water (10 °C) were conducted. The results show that Chemical Oxygen Demand (C.O.D.) values are reduced by 84% (ozone treatment) against 51% (hot water treatment). Furthermore, many researchers have evaluated the effectiveness of ozone against microorganisms that can colonize metal surfaces.

Greene et al. [89] studied the effect of ozonated deionized water against psychrophilic contaminating microflora (*Pseudomonas fluorescens* and *Alcaligenes faecalis*) on stainless steel plates, showing that a 10-min exposure with a concentration of 0.5 ppm decreased the microbial growth by more than 4 log_10_. For the experimentation, stainless steel plates were incubated in UHT pasteurized milk and inoculated with pure cultures of *Pseudomonas fluorescens* (ATCC 949) or *Alcaligenes faecalis* (ATCC 337). Since these are metal surfaces, the corrosive power of ozone must always be considered. The use of ozonated water is recommended, instead of hot water and chlorine, when the surfaces of the milk processing equipment are not damaged. Recently, the potential of disinfection effectiveness of single and synergistic ozone (10 ppm for 15 min) and UVC (1 cm or 15.56 mW cm^−2^ for 15 min) treatment for the sterilization of bacteria and fungi (*Escherichia coli*, *Staphylococcus aureus*, *Candida albicans*, and *Aspergillus fumigatus*) was studied on different material surfaces (stainless steel, polymethyl methacrylate, copper, surgical facemask, denim, and a cotton-polyester fabric) [90].

In the work of Dosti et al. [91], fresh 24-h bacterial cultures were treated with ozone (0.6 ppm for 1 min and 10 min), chlorine (100 ppm for 2 min), or heat (77 ± 1 °C for 5 min), showing its effectiveness against food spoilage microorganism in synthetic broth. The bacterial biofilm on the metal coupons was significantly reduced by ozone and chlorine, but with no significant difference between ozone and chlorine, except for *P. putida* (ozone was more effective than chlorine).

Greene et al. [47] observed that 0.4–0.5 ppm of ozone, pulsed into water at 21–23 °C for 20 min per day over a 7-day period, caused a certain degree of weight loss of all materials tested (i.e., aluminum, copper, stainless steel, and carbon steel), but only weight loss for carbon steel was significant. Therefore, special attention is required when the treatment is used for dairy chilling water systems with copper or carbon steel parts.

Megahed et al. [92,93] studied the effect of gaseous-ozone and aqueous-ozone (from 1 to 10 ppm) in commonly used devices of the dairy industry (plastic, nylon, rubber, and wood) contaminated by cattle manure-based pathogens. It was observed that aqueous-ozone treatment at a concentration of 4 ppm or greater reduced bacterial load below detectable limits within 2 min of exposure predominantly on the plastic surface, while other surfaces were largely decontaminated after 4 min of treatment. Gaseous O_3_ cannot be an alternative to aqueous O_3_ in reducing the manure-based pathogens to a safe level, particularly in complex environments. The results obtained are in line with different previous studies [94].

It is important to consider the contamination of surfaces by spoilage bacteria, like *Pseudomonas* spp. and pathogenic bacteria, like *Listeria monocytogenes* and *Salmonella* spp., that lead to recurrent food contamination, with problems related to the shelf life and safety of dairy products.

Biofilm development in food processing facilities is prevented by using common chemical sanitizers, but their use has some disadvantages, such as damaging environmental impacts and harmful consequences for human health [52]. The use of ozone could be a promising technique to prevent spoilage or pathogenic bacteria contamination [22].

Shelobolina et al. [95] studied the effect of dissolved ozone (5 ppm for 20 min) on *Pseudomonas* biofilm on various surfaces, and they observed that ozone can be effective. Biofilms on plastic materials showed inactivation effects, such as for glass, while biofilms grown on ceramics were more difficult to inactivate. Therefore, it is important to use non-porous materials in industrial and clinical settings. It has also been demonstrated that ozone can have an antimicrobial effect in association with other technologies: for example, ozone water and hydrogen peroxide solution were effective on *P. fluorescens* biofilm.

A sequential treatment with 1.0 and 1.7 mg L^−1^ of ozone, followed by 0.8 and 1.1% of hydrogen peroxide, showed synergistic disinfection effects [96].

In a study of a cheese production plant, the gaseous ozonation of 2 ppm was provided during a weekend for 15 min (first treatment) and for 120 min (second treatment), when staff were not present. Testing for *L. monocytogenes* was carried out for a total of 360 environmental samples, over a period of 12 months, in 15 areas before and 15 areas after ozonation: there was a significant reduction in *L. monocytogenes* isolations from 15.0% in pre-zoning samples to 1.67% in post-ozonation samples in all areas, to include the ozonation regime in the hygiene-health program. No negative effects of ozonation treatment were noted on the surfaces and equipment [97].

Instead, Shao et al. [98] studied the effect of ozone water on mature *S. aureus* and *Salmonella* spp. biofilm on stainless steel surfaces, using acidic electrolyzed water, ozone water, or ultrasound (40 kHz) alone, and combinations of ultrasound and disinfectants. They detected less than 0.8 Log CFU (colony-forming units) cm^−2^ of cells reduction in biofilm exposed to ozonized water (16 mg L^−1^) for 20 min. Only in a few studies was the efficacy of ozone explored against biofilms formed by bacteria belonging to *the Salmonella* genus [99].

Indoor air in the dairy industry constitutes a source or a vehicle of microbial contamination, causing food safety and product shelf-life problems. Masotti et al. [100] monitored the air microbial load in the dairy plant and evaluated the impact of air disinfection through ozonation or chemical aerosolization by hydrogen peroxide. Ozonated air had a constant flow rate of 40 L min^−1^, and it generated 1.5 g ozone per hour (11–12 p.m. and 1–3 a.m. from Friday to Sunday). Hydrogen peroxide aerosolization was realized producing particles in the range of 5–15 μm at a concentration of 5–15% for 16 and 20 min. Both techniques were effective against airborne microorganisms.

**Table 3 foods-12-00987-t003:** Ozone sanitization in working surface areas and equipment.

Area	Treatments	Target	Result	References
Ozone sanitization in working surface areas and equipment	Ozonated cold water (10 °C) for 15 min	Ozone treatment on stainless steel surfaces to remove milk residues	Chemical Oxygen Demand values are reduced by 84%	Guzel-Seydim et al., 2000 [88]
	10 min exposure with 0.5 ppm of ozonated deionized water	Effectiveness of ozone against microorganisms that can colonize metal surfaces	Microbial growth decreased by more than 4 log_10_	Greene et al., 1993 [89]
	Fresh 24-h bacterial cultures were exposed to ozone (0.6 ppm for 1 min and 10 min), chlorine (100 ppm for 2 min) or heat (77 ± 1 °C for 5 min)	Ozone, chlorine and heat applications were compared for killing effectiveness against food spoilage bacteria in synthetic broth	Ozone and chlorine significantly reduced the biofilm bacteria adhered to the metal coupons as compared to the control. No difference between ozone and chlorine inactivation of the bacteria. Ozone killed *P. putida* more effectively than chlorine	Dosti et al., 2005 [91]
	20 min exposure with 0.4–0.5 ppm of ozonated water at 21–23 °C for 7 days	Effect of ozone on more materials	Weight loss of all materials tested, but only weight loss for carbon steel was significantly	Greene et al., 1999 [47]
Ozone sanitization in working surface areas and equipment	Ozone concentrations from 1 to 10 ppm for 4 min	Effect of gaseous-ozone and aqueous-ozone in commonly used devices of the dairy industry	All surfaces were largely decontaminated after 4 min treatment	Megahed et al., 2018, 2019 [92,93]
	Ozone concentrations between 300 and 1500 ppm for 10–480 s	Effect of gaseous ozone and aqueous ozone in commonly used devices of the dairy industry	*Escherichia coli* and *Staphylococcus aureus* died for 99.99%	Kowalski et al., 1998 [94]
	*Pseudomonas fluorescens, Staphylococcus aureus*, and *Listeria monocytogenes* were treated with ozonized water (0.5 ppm) by immersion in static condition, ozonized water under flow conditions, and gaseous ozone at different concentrations (0.1–20 ppm) (20–60 min)	Inactivation of foodborne bacteria biofilms by aqueous and gaseous ozone	Aqueous ozone under static conditions resulted in an estimated viability reduction of 1.61–2.14 Log CFU cm^−2^ after 20 min, while reduction values were higher (3.26–5.23 Log CFU cm^−2^) for biofilms treated in dynamic conditions. With gaseous ozone, the highest concentrations estimated a complete inactivation	Marino et al., 2018 [52]
	Ozone concentration of 5 ppm for 20 min	Effect of dissolved ozone on *Pseudomonas biofilm* on various surfaces	The biofilms growth on the different materials were inactivated	Shelobolina et al., 2018 [95]
Ozone sanitization in working surface areas and equipment	Ozone concentration of 1.0–1.7 mg L^−1^ followed by 0.8–1.1% of hydrogen peroxide	Antimicrobial effect of ozone water in combination with a hydrogen peroxide solution	Synergic treatments showed an antimicrobial effect against *Pseudomonas fluorescens* biofilm	Tachikawa et al., 2014 [96]
	Ozone concentration of 2 ppm for 15 min Monday to Friday and 120 min Saturday and Sunday	Reduce the environmental colonization of *Listeria monocytogenes* by means of an ozonation regime in all production areas	It showed a reduction in *Listeria monocytogenes* isolations from 15.0% in pre-zoning samples to 1.67% in post-ozonation samples in all areas	Eglezos and Dykes, 2018 [97]
	Ozonated water with 16 mg L^−1^ of ozone for 20 min	Effect of acidic electrolyzed water, ozone water, or ultrasound on *Staphylococcus aureus* and *Salmonella* spp. biofilm on stainless steel surfaces	Less than 0.8 Log CFU cm^−2^ of cells reduction in biofilm exposed to ozonized water	Shao et al., 2020 [98]
	Ozonated water with an ozone concentration of 1 ppm and 1–2% of malic acid for 20–40 min	The combined effect of malic acid and ozone as sanitizer to inhibit the biofilm formation by *Salmonella typhimurium* on different food contact surfaces	Combination of malic acid with ozone reduced the biofilm formation on plastic bags, as well as on PVC pipes, suggesting it as an effective disinfectant for food contact surfaces	Singla et al., 2014 [99]
Ozone sanitization in working surface areas and equipment	Ozonated air had constant flow rate of 40 L min^−1^ and it generated 1.5 g h^−1^ of ozone (11–12 p.m. and 1–3 a.m. from Friday to Sunday). Hydrogen peroxide aerosolization was realized producing particles in the range of 5–15 μm at a concentration of 5–15% for 16–20 min.	Effect of ozonation or chemical aerosolization through hydrogen peroxide to monitor the air microbial load in the dairy factory and in evaluating the air disinfection	Ozonation and hydrogen peroxide aerosolization were effective techniques in the inactivation of airborne microorganisms	Masotti et al., 2020 [100]

### 3.4. Ozone Treatment in Milk Production

In the treatment of raw milk, the use of ozone is debated: in fact, in contrast to the traditional thermal pasteurization process, ozone treatment minimizes the loss of the nutritional properties of milk, but it does not always show a total inactivating capacity on microorganisms [101] (Table 4).

Torlak and Sert [102] studied ozone treatment to eliminate *Cronobacter sakazaki* (responsible for fatal infections in infants) from milk powder. They treated both whole and skimmed milk with different results: for the skimmed milk, the initial levels of *Cronobacter* were reduced by 2.71 and 3.28 log, following an exposure of 120 min at the concentration of 2.8 mg L^−1^ and 5.3 mg L^−1^, and the whole milk sample showed a smaller logarithmic reduction.

Rojek et al. [103] used pressurized ozone (5–35 mg L^−1^ for 5–25 min) to safeguard skim milk by diminishing its microbial masses, and this treatment looked foremost to decrease the number of psychrotrophs by more than 99%.

Sheelamary and Muthukumar [104] totally wiped out *Listeria monocytogenes* from both crude and marked milk tests through gaseous ozonation, with a mean viable count of 5.5 and 5.7 log_10_ CFU mL^−1^, respectively. The results showed complete removal of *Listeria monocytogenes* after 15 min of treatments with a controlled flow rate of 0.5 m L^−1^ of oxygen (0.2 g h^−1^ of ozone).

Cavalcante et al. [101] evaluated in crude milk that ozone gas at 1.5 mg L^−1^ for 15 min was found to decrease bacterial as *Listeria monocytogenes*, *Enterobacteriaceae*, and *Staphylococcus*, and contagious checks by up to 1 log_10_ cycle.

A study by Alsager et al. [105] analyzed the decomposition of various antibiotic compounds in milk samples by the ozonation process. They observed that increasing concentration of gas ozone (75 mg L^−1^) caused a 95% reduction in antibiotics in milk samples.

This was confirmed by Liu et al. [106]: antibiotic residues in pasteurized milk reach tolerance levels by O_3_ in a vortex reactor (2.16, 4.54, and 6.12 mg h^−1^), and no antimicrobial activity was detected in the milk treated. The results showed that post-ozonation of 400 mL of 5.52 μM various antibiotics for 20–40 min, their residue concentrations (50.8–84.1 μg L^−1^) satisfy the relevant maximum residue limits. Therefore, antibiotic residues in milk arrive at tolerance levels by O_3_ in a vortex reactor. Increasing the input O_3_ concentration, O_3_ flow rate, and temperature can accelerate the degradation and shorten the time to meet the relevant MRLs.

de Oliveira Souza et al. [107] studied the effect of gaseous ozone on the inactivation of *Escherichia coli* O157:H7 inoculated on an organic substrate and the efficacy of ozonated water in controlling the pathogen. It was inoculated in milk with different compositions: unhomogenized whole milk (UHWM), homogenized whole milk (HWM), skim milk (SM), lactose-free skim milk (LFSM), and lactose-free whole milk (LFWM), which were sterilized by boiling in the laboratory to guarantee the elimination of microorganisms and cooled before inoculation. Milk was utilized only as a substrate, contemplating the differences in fat and lactose contents. The same procedures were also conducted in distilled water for comparison. *Escherichia coli* O157:H7 was inoculated in different ozonated water treatments (35 and 45 mg L^−1^ for 0, 5, 15, and 25 min). In a second experiment, the water was ozonated at 45 mg L^−1^ for 15 min. *E. coli* O157:H7 was exposed for 5 min to the ozonated water immediately after the treatment, and after storage for 0.5, 1.0, 1.5, 3.0, and 24 h at 8 °C. The results showed that in lactose-free homogenized skim milk, was observed a reduction of 1.5 log cycles for ozonation periods of 25 min at the concentrations tested. Overall, ozonated water was effective in all treatments, but the efficiency of ozone is influenced by the composition of the organic substrates. The refrigerated ozonated water put in storage for up to 24 h was effective to control *E. coli* O157:H7.

The degradation of mycotoxins and aflatoxins is also possible through the use of ozone [108]. As reported by Ismail et al. [109] when milk samples were exposed to gas ozone (80 mg min^−1^ for 5 min), and aflatoxins were reduced by 50%.

Khoori et al. [110] studied the synergistic effect of ozonation (9.99 mg min^−1^), UV light radiation (4.99 J cm^−2^), and pulsed electric field processes (13.15 µs) on the decrease of aflatoxin in probiotic milk. Data showed an effect in reducing the levels of aflatoxins (total and M1) in acidophilus milk samples. Additionally, the bacterial viability was reduced through these methods, with a level of *Lactobacillus acidophilus* in the final product of 10^6^ CFU g^−1^.

Sert and Mercan [111] studied the effect of ozone treatment (OT) against milk concentrate (MC) and whey concentrate (WC), evaluating the physicochemical and textural properties and the effect on the microbial, antibiotic, and aflatoxin content. The ozone was applied to MC and WC samples for 0 (control), 5, 10, 15, 30, and 60 min. The treatment was performed as previously described [112], with some changes. Ozone treatment was conducted at 20 ± 2 °C. An ozone generator with a capacity of 3.5 g ozone h^−1^ was used. The results showed that OT (60 min) reached 18.9 and 9.9% degradation of the aflatoxin M1 in MC and WC, respectively. It caused the degradation of β–lactam and tetracycline in concentrates. The volume mean diameter of MC was increased by the ozone, while for WC it was decreased. In both MC and WC samples, the ozone treatment was successful in decreasing the microbial load. For MC samples, the coliform decreased from 3.19 ± 0.04 log CFU g^−1^ at time 0 to 2.37 ± 0.06 log CFU g^−1^ at time 60. For WC samples this decreased from 2.79 ± 0.06 log CFU g^−1^ to 2.04 ± 0.06 log CFU g^−1^. For MC samples, *Enterobacteriaceae* decreased from 3.33 ± 0.04 log CFU g^−1^ at time 0 to 2.60 ± 0.05 log CFU g^−1^ at time 60. For WC samples, this decreased from 3.17 ± 0.05 log CFU g^−1^ to 2.54 ± 0.06 log CFU g^−1^. For MC samples, the staphylococci decreased from 2.65 ± 0.08 log CFU g^−1^ at time 0 to 1.89 ± 0.06 log CFU g^−1^ at time 60. For WC samples, this decreased from 2.52 ± 0.03 log CFU g^−1^ to non-detectable. For MC samples, the total mesophilic aerobic bacteria decreased from 3.39 ± 0.08 log CFU g^−1^ at time 0 to 2.93 ± 0.08 log CFU g^−1^ at time 60. For WC samples, this decreased from 2.93 ± 0.06 log CFU g^−1^ to 2.35 ± 0.07 log CFU g^−1^. For MC samples, the yeast and mold decreased from 2.75 ± 0.07 log CFU g^−1^ at time 0 to non-detectable at time 60. For WC samples from 2.36 ± 0.06 log CFU g^−1^ to non-detectable (Figure 3).

**Table 4 foods-12-00987-t004:** Ozone milk treatment.

Area	Treatments	Target	Result	References
Ozone milk treatment	Ozone concentration of 2.8 mg L^−1^ and 5.3 mg L^−1^ for 120 min	Ozone treatment to eliminate *Cronobacter sakazaki* from skimmed milk powder	Initial levels of *Cronobacter* were reduced by 2.71 and 3.28 log	Torlak and Sert, 2013 [102]
	Pressurized ozone concentration of 5–35 mg L^−1^ for 5–25 min	Safeguard skim milk by diminishing its microbial masses	Pressurized ozone decrease the number of psychrotrophs by more than 99%	Rojek et al., 1995 [103]
	Controlled flow rate 0.5 m L^−1^ of oxygen (0.2 g h^−1^ of ozone) for 15 min	Gaseous ozonation treatment to reduce *Listeria monocytogenes* from both crude and marked milk tests	Gaseous ozonation totally wiped out *Listeria monocytogenes* from both crude and marked milk tests with a mean viable count of 5.5 and 5.7 log_10_ CFU mL^−1^, respectively.	Sheelamary and Muthukumar, 2011 [104]
	Ozone gas at 1.5 mg L^−1^ for 15 min	Ozone gas bubbling effects in the quality of raw milk	Ozone decreases bacterial and contagious checks by up to 1 log_10_ cycle	Cavalcante et al., 2021 [101]
	Gas ozone concentration of 75 mg L^−1^ for 10 min	Decomposition of various antibiotic compounds in milk samples by ozonation process	Gas ozone caused a 95% reduction in antibiotics in milk samples	Alsager et al., 2018 [105]
Ozone milk treatment	Ozone concentrations of 2.16, 4.54, and 6.12 mg h^−1^ at selected volumes (1 L of each) and times of 2, 4, and 6 h	Use of ozonation in a vortex reactor for removing antibiotics from milk	Post-ozonation of 400 mL of 5.52 μM various antibiotics for 20–40 min, their residue concentrations were 50.8–84.1 μg L^−1^ satisfying the relevant maximum residue limits	Liu et al., 2022 [106]
	In the first experiment, ozone concentrations were 35 and 45 mg L^−1^ for 0, 5, 15, and 25 min. In a second experiment, the water was ozonated at 45 mg L^−1^ for 15 min	Efficacy of ozonated water in controlling *Escherichia coli* O157:H7	Reduction of 1.5 log cycles for 25 min in lactose-free homogenized skim milk. The refrigerated ozonated water put in storage for up to 24 h was effective to control *E. coli*	de Oliveira Souza et al., 2019 [107]
	Gas ozone concentration of 80 mg min^−1^ for 5 min	Degradation of aflatoxins on milk samples	Aflatoxins were reduced by 50%	Ismail et al., 2018 [109]
	Ozonation (9.99 mg min^−1^), UV light radiation (4.99 J cm^−2^) and pulsed electric field processes (13.15 µs)	Synergistic effect of ozonation, UV and pulsed electric field processes on the decrease of aflatoxin in probiotic milk	Reduction of levels of aflatoxins (total and M1). *Lactobacillus acidophilus* in the final product of 10^6^ CFU g^−1^	Khoori et al., 2020 [110]
Ozone milk treatment	3.5 g ozone h^−1^ at 20 ± 2 °C for 60 min	Effect of ozone treatment against milk concentrate and whey concentrate, evaluating the physicochemical and textural properties and the effect on the microbial, antibiotic, and aflatoxin content	Ozone treatment reached 18.9% and 9.9% degradation of the aflatoxin M1 in milk concentrate and whey concentrate, respectively	Sert and Mercan, 2021 [111]

### 3.5. Ozone Technology for Dairy Wastewater Care

In the dairy industry, water is used in a wide range of uses, generating significant volumes of wastewater rich in organic matter. It is currently purified with physicochemical and biological methods. Many studies have evaluated the possibility of using ozone for their treatment (Table 5), with the aim of recovering and reusing water [113,114,115,116].

Làszlò et al. [117] confirmed the ability of ozone (30 mg dm^−3^) to reduce the content of organic pollutants in wastewater at 25 °C for 5 min; thanks to its microflocculation effect, in fact, the effectiveness of the following nanofiltration phase increases and, consequently, the reduction of COD is performed; in addition, there is a 40% increase in the biodegradability of nanofiltration residues.

Additionally, dos Santos Pereira et al. [118] investigated the degradation of organic matter in a synthetic dairy wastewater. In this study, dairy wastewater has a chemical oxygen demand of 2.3 g L^−1^, and they treated this wastewater through an oxidation process using ozonation combined with hydrogen peroxide (30% *w*/*w*, [H_2_O_2_] = 9.007 mol L^−1^ and density of 1.1 g mL^−1^) and catalyzed by manganese (Mn^2+^) in alkaline conditions (MnSO_4_.H_2_O, 98% of purity). It was observed that the optimal condition for the ozonation catalyzed by manganese at alkaline medium (chemical oxygen demand removal of 69.4%) can be obtained at pH 10.2 and Mn^2+^ concentration of 1.71 g L^−1^, with COD removals above 60%.

In the work of Li et al. [119], it was observed that Mn-Fe-Ce/γ-Al2O_3_ can be used for catalytic ozonation of dairy farm wastewater, with efficient treatment results. The biodegradability of wastewater was drastically improved after 20 min of reaction time with Mn-Fe-Ce/γ-Al_2_O_3_ catalyst, which has a dosage of 12.5 mg·L^−1^·min^−1^ of ozone. The pH value was 9 and the catalyst dosage was 15 g·L^−1^. The mechanism involves the conversion of refractory color-developing organics into small-molecule organics that can be biodegraded. The results showed that COD removal ratio of dairy farming wastewater can reach 48.9%. The BOD/COD increased from 0.21 to 0.54. Therefore, it has been observed that as the catalyst dosage increases (0–25 g/L), the chemical oxygen demand decreases.

Recently, Zhu et al. [120] studied the effect of ozone on undiluted dairy farm liquid digestate that contains high levels of organic matters, chromaticity, and total ammonia nitrogen (TAN), resulting in inhibition of microalgal growth. Ozone greatly promoted microalgal growth, but it did not clearly reduce pollutants. After cascade pretreatment, TAN, total nitrogen (TN), COD, and chromaticity were reduced by 80.2%, 75.4%, 20.6%, and 75.8%, respectively.

In the work of Chen et al. [121], dairy wastewater was pre-treated with ultrasound (US 200 W), ozone (4.2 mg O_3_ L^−1^), and US combined with ozone (US/ozone) to study the fate of enteric indicator bacteria and antibiotic resistance genes (ARGs), and anaerobic digestion (AD). All pre-treatments were performed for 10 min, 20 min, and 30 min, respectively. US/ozone pre-treatment was effective in the inactivation of enteric indicator bacteria. Total coliforms and enterococci were reduced by 99% and 92% after 30 min US/ozone pre-treatment. Together, pre-treatment and AD clearly inhibited the enrichment of ARGs in relative abundance.

Chang et al. [122] designed a continuous type O_3_ treatment system (43.26, 87.40, and 132.46 mg L^−1^) to eliminate pathogens such as *Salmonella Typhimurium* and *Escherichia coli* O157: H7 in liquid dairy waste, and they observed that the ozone treatment reduced the amount of *E. coli* O157: H7 and of *S. typhimurium*, and the reductions increased with the exposure time, particularly at 87.40 or 132.46 mg·L^−1^. These results were confirmed by Choi et al. [123].

### 3.6. Use of Ozone Technology in Cheese-Making Processes and in Storage/Maturation Condition

Many studies demonstrate the effectiveness of ozone in preventing the growth of mold in maturing environments against cheese mites (Table 6).

Gibson et al. [124] demonstrated the bacteriostatic effect by applying two different concentrations of ozone in the ripening phase of Cheddar cheese, verifying its effectiveness both at concentrations of 3–10 ppm for 30 days and at those of 0.2–0.3 ppm for 63 days with a difference of 6 percentage points in favor of the highest concentrations. Gabriel ‘yants’ et al. [125] confirmed these observations: the application of ozone under refrigerated conditions on Russian and Swiss-type cheese prevented mold growth for four months without damaging the sensory properties and chemical composition, while growth was observed on the control sample already after one month of storage. They stored the cheeses under refrigeration (2–4 °C, 85–90% RH) with or without ozonation of the air in the room. Periodic treatments were undertaken with concentrations of 2.5–3.5 ppm of gaseous ozone for 4 h at 2- to 3-day intervals, preventing mold growth on packaging materials for up to 4 months.

Other studies verify the effectiveness of ozone against the bacterial population: Morandi et al. [71] applied 4 ppm of gaseous ozone for 8 min at different stages of maturation and showed the effectiveness in controlling *Listeria monocytogenes* (artificially surface-inoculated up to 10^3^ CFU g^−1^) on Ricotta Salata di Pecora (below 10 CFU g^−1^) and, limited to the first days of maturation on Gorgonzola PDO (Protected Designation of Origin) and Taleggio PDO, the effect was a complete elimination; Cavalcante et al. [126] proposed treatments with ozonated water (2 mg L^−1^ for 1–2 min) for washing Minas Frescal cheese during storage, highlighting, once again, the ability to reduce the initial microbial load (by approximately 2 log_10_ cycles) without damaging the sensory and physical–chemical properties of the product. However, the treatment did not show efficacy in controlling the growth rate of the surviving microflora.

Serra et al. [127] have shown the effectiveness of ozone treatment against the fungal spores that colonize the maturing rooms, but they have also shown that the treatment may not eliminate the molds present on the surface of the cheeses. They studied the effect of ozone to reduce molds in a cheese-ripening room (3500 m^3^) with a temperature of 5 ± 1 °C and a relative humidity of >80%. The efficacy of this treatment was evaluated in air and on surfaces through sampling on a weekly basis over a period of 3 months. The results demonstrated that ozonation decreased the viable airborne mold load, but not on surfaces, resulting in a 10-fold reduction compared with the level observed for the control. The mold genera usually isolated in the air were *Penicillium*, *Cladosporium*, and *Aspergillus* (89.9% of the mold isolates). They used 8 g h^−1^ as the rate of ozone generation.

In their study, Guzzon et al. [128] explored the microbiota of the red–brown defect in smear-ripened cheese and how to prevent it using different cleaning systems. Red–brown pigmentation can occasionally form in this type of cheese, such as Fontina, during the ripening process due to an over-development of the typical microbiota present on the rind. The microbiota of the shelf had a role in this defect in cheese. Different systems were tested for cleaning the wooden shelves (WS): washing with hot water and ozone treatment. The results showed that *Actinobacteria*, dominant on the WS, indicated to be responsible for the red–brown pigmentation; they were also in traces in the defected samples. *Galactomyces* and *Debaryomyces* were the main species for the yeast population, with *Debaryomyces* as the most dominant species on the shelves used during the maturation of the red–brown defective cheese. Even if the hot water treatment decreased the microbial load of shelves, only the use of ozone guaranteed a total elimination of yeast and bacteria, with no red–brown defect on the cheese rind. The ozone treatment (OT) was done using ozone in two different forms: gaseous (dry) or water-dissolved (wet). Therefore, washing plus ozone treatment ensures that the WS will be safe for the next ripening process. After the study, some trained experts reported that the wheels ripened on shelves cleaned by wet ozone treatment did not have the red–brown defect and no difference in cheese color, taste, and texture.

In the study of Alexopoulos et al. [129], an ozone stream (2.5–3 ppm) for 0, 10, 30, and 60 s was used on the surface of freshly filled yogurt cups (240 g) before storage for the development of the curd (24 h) to prevent cross-contamination from spoilage airborne microorganisms. Additionally, the brine solution was bubbled with ozone at different times and applied for the ripening of white (feta type-400 g) cheese. Ozone gas was supplied for 0, 10, 20, and 30 min to the brine, where the cheese sample was then left for ripening for a period of 2 months. Products were monitored for microbial load and tested for their sensorial characteristics. In ozonated yogurt samples, data showed a reduction in mold counts of about 0.6 Log CFU g^−1^ (25.1%) by the end of the monitoring period against the control samples. In white cheese matured with ozonated brine (1.3 mg L^−1^ O_3_, NaCl 5%), the treatment for two months decreased some of the mold load without offering advantages over the use of traditional brine (NaCl 7%). However, some alterations in the sensorial quality were observed, perhaps due to the organic load of the brine that soon neutralizes ozone. For the cheese samples ripened in 60 min ozonated brine, there was a reduction in the attribute of flavor, according to the global acceptance. Therefore, factors of time and concentration of ozone are essential, and they must be configured.

The efficacy of ozone is non-selective towards harmful or useful bacteria. It becomes challenging when lactic acid bacteria (LAB—one of the most significant groups of probiotic organisms), commonly used in fermented dairy products, is destroyed by the application of ozone. In this work, LAB was present in all yogurt samples in an adequate number (*S*. *thermophillus* 9.1 ± 0.5 Log CFU g^−1^ and *L*. *delbrueckii* ssp. *bulgaricus* 8.8 ± 0.6 Log CFU g^−1^). Although a drastic decline was observed after 50 days of storage by the end of the monitoring time, nevertheless no statistical differences were observed between the control and the ozonated samples. Therefore, ozone did not affect good microflora, maintaining the functional character of the yogurt, with LAB values higher than 7 Log CFU g^−1^ for a period of 70 days.

Segat et al. [130] used ozone treatment to reduce the microbial spoilage load on “mozzarella” cheese, concluding that the cooling water treatment (15 °C) with ozone (2 mg L^−1^) improves the microbiological qualities, positively affecting the shelf-life of the products. The results showed that “mozzarella” cheese samples that were cooled in water and pretreated were characterized by low microbial counts as compared to control samples cooled with non-ozonated water, following 21 d of storage (by 3.58 and 6.09 log_10_ CFU g^−1^ lower total plate counts and *Pseudomonas* spp. counts, respectively, compared to control samples).

Sert et al. [112] showed the effect of ozone on butter samples in their work, considering that the treatment can be applied to produce butter as a non-thermal technology. Raw cream was ozone-treated for 5 (OT-5), 15 (OT-15), 30 (OT-30), and 60 (OT-60) min. The ozone application was conducted at 8 ± 2 °C and, after the treatment, the churning was carried out at 4 ± 2 °C. Then, the samples were packaged and stored at 4 °C for 24 h. The ozone treatment increased the firmness, consistency, and fat particle size values of samples. Instead, it decreased the color parameter b value, so the yellowness of butter. OT-60 caused 2.01 log reduction in Staphylococci, and over 15 min, the ozone completely inactivated Salmonella and yeast-mold, while coliform was not detected in OT-30 and OT-60. The results showed a higher microbiological quality in butter from raw cream with ozone treatment. Instead, OT decreased the oxidative stability of butter and up to 15 min increased the spreadability characteristic of the sample.

Panebianco et al. [131] studied the effect of ozone on *Listeria monocytogenes* contamination and the resident microbiota on the Gorgonzola cheese rind. Concentrations of 2 and 4 ppm for 10 min were used against *L. monocytogenes* and resident microbiota of cheese rind samples stored at 4 °C for 63 days. Results showed that in ozonized rinds the final loads of *L. monocytogenes* were ~1 log CFU g^−1^ higher than controls. No significant differences were found for the other microbial determinations and the resident microbiota between ozonated and control samples.

Recently, Tabla and Roa [132] studied the effect of gaseous ozone in soft cheese ripening, in particular its effect on the rind microorganisms, evaluating the sensorial quality. The experiment was conducted at different production scales. On a pilot plant scale, the samples were inoculated with *Mucor plumbeus, Kluyveromyces marxianus*, and *Pseudomonas fluorescens*, while on an industrial scale they evaluated the effect of ozone on naturally present microorganisms. For the experiment at pilot plant scale, cheeses were surface-inoculated after salting to a final concentration of 10^5^ spores g^−1^ of rind for *Mucor plumbeus* and 10^7^ CFU g^−1^ of rind for *Kluyveromyces marxianus* and *Pseudomonas fluorescens*. The ripening was conducted in 8 m^3^ ripening rooms for 30 days with gaseous ozone at concentrations of 0, 2, and 6 mg m^−3^. For the experiment at dairy plant scale, a total of 30 cheeses with 15 days of ripening were subjected to maturation with gas ozone at concentrations of 2 mg m^−3^ for 45 days. The results showed that gaseous ozone treatments had a significant fungistatic effect on cheese surface molds for pilot and industrial scale, but its effect on yeast growth was only observed at pilot plant scale, and for *Pseudomonas* ssp., the ozone concentrations used did not affect it. For the sensory quality of the cheese, its typicality was not influenced by gas ozone, and the ozonized cheeses had significantly higher scores than the control samples for the appearance of the rind and color, preventing cheese rind discoloration.

In ripened cheeses, an important problem is the presence of mites that live and grow on the rind of the product. Some studies have shown that the most represented species in the dairy industry are *Acarus siro*, *Tyrophagus casei*, *T. longior*, *T. palmarum*, and *T. putrescentiae* [36,37,38]. The ripening environment is optimal for the proliferation of mites, above all by the high relative humidity and temperatures that hover around 12–13 °C; it has, in fact, been seen that at temperatures of °C these parasites do not survive, but there is also an excessive prolongation of the ripening times which makes them inapplicable [133], and in most cases the presence of mites in environments of maturation is associated with important economic losses. The intake of mites can cause the appearance of symptoms ranging from dermatitis to allergic rhinitis and asthma, up to gastro-intestinal problems. It has been seen that water at a temperature higher than 71.1 °C [36] eliminates the mites in a few seconds; moreover, there are several effective chemical treatments, and the most used substances are organophosphate compounds. However, chemical products leave residues and, in some cases, such as that of methyl bromide, they have been banned, as they have harmful effects on human health. For this reason, it is essential that research identifies alternative control methods. Therefore, the interest in the methods of sanitizing the maturing environments with ozone is growing. The effectiveness of ozone in killing mites has been ascertained thanks to a few studies, even in dairy environments. For example, Pirani [59] verified the effectiveness of ozone treatment for the control of mite infestations on matured speck products.

In a new study, Pecorino cheese samples were treated for 150 days overnight with gaseous ozone (200 and 300 ppb, 12 °C, and 85% R.H. for 8 h per day). It was observed that, starting from 25 days of storage, 200 ppb of ozone reduced the growth of mites and significantly reduced bacteria, molds, and yeast count, starting from 75 days of storage. Furthermore, it was observed that no significant differences were shown between the control samples and ozone treatment at 200 ppb for the centesimal composition of Pecorino cheese. However, treatments with ozone 300 ppb contained microbiological and mite growth but did not have the same positive impact on some aspects of overall quality [134].

**Table 6 foods-12-00987-t006:** Use of ozone technology in cheese making processes and in storage/maturation conditions.

Area	Treatments	Target	Result	References
Use of ozone technology in cheese making processes and in storage/maturation condition	Ozone concentrations of 3–10 ppm for 30 days and of 0.2–0.3 ppm for 63 days	Effect of ozone in the ripening phase of Cheddar cheese	Higher ozone concentrations showed 6% more bacteriostatic action than lower concentrations	Gibson et al., 1960 [124]
	Cheese refrigeration at 2–4 °C, 85–90% RH with or without ozonation of the air in the room. Periodical treatments were done with concentrations of 2.5–3.5 ppm of gas ozone for 4 h at 2- to 3-day intervals	Application of ozone under refrigerated conditions on Russian and Swiss-type cheese to prevent mold growth	Ozone prevented mold growth for four months, while growth was observed on the control sample already after one month of storage. It prevented also mold growth on packaging materials for up to 4 months	Gabriel ’yants’ et al., 1980 [125]
	4 ppm of gaseous ozone for 8 min at different stages of maturation	Efficacy of ozone against *Listeria monocytogenes* (artificially surface-inoculated up to 10^3^ CFU g^−1^)	On Ricotta Salata below 10 CFU g^−1^ and, limited to the first days of maturation on Gorgonzola PDO and Taleggio PDO, the effect was a complete elimination	Morandi et al., 2009 [71]
	Ozonated water (2 mg L^−1^ for 1–2 min)	Ozonized water treatments for washing Minas Frescal cheese during storage	Reduction of the initial microbial load (by 2 log_10_ cycles)	Cavalcante et al., 2013 [126]
Use of ozone technology in cheese making processes and in storage/maturation condition	Cheese preserved with a temperature of 5 ± 1 °C and a relative humidity of >80% for 3 months. It was used 8 g h^−1^ as the rate of ozone generation.	Efficacy of ozone treatment against molds present on the surface of cheeses and of the fungal spores that colonize the maturing rooms	Ozonation reduced the viable airborne mold load, but not on surfaces, resulting in a 10-fold reduction compared with the level observed for the control	Serra et al., 2003 [127]
	Gaseous ozone was produced of 40 g h^−1^ instead ozone water was done by coupling the same generator of ozone to a water bath (nominal volume of 100 L)	Explore the microbiota of the red–brown defect in smear-ripened cheese and how to prevent it	Ozone treatment guaranteed a complete elimination of yeast and bacteria, with no red–brown defect on the cheese rind	Guzzon et al., 2015 [128]
Use of ozone technology in cheese making processes and in storage/maturation condition	Ozone stream (2.5–3 ppm) for 0, 10, 30, and 60 s was used on the surface of freshly filled yogurt cups (240 g) before storage for the development of the curd (24 h). The brine solution bubbled with ozone was applied for the ripening of white (feta type, 400 g) cheese. Ozone gas was supplied for 0, 10, 20, and 30 min to the brine where cheese sample was then left for ripening (2 months)	Prevent contamination from spoilage airborne microorganisms	In ozonated yogurt samples, data showed a reduction in mold counts of about 0.6 Log CFU g^−1^ (25.1%) by the end of the monitoring period against the control samples. In white cheese matured with ozonated brine (1.3 mg L^-1^ O_3_, NaCl 5%) the treatment during two months decreased some of the mold load	Alexopoulos et al., 2017 [129]
	Cooling water (15 °C) with ozone (2 mg L^−1^)	Reduce the microbial spoilage load on Mozzarella cheese	Samples cooled in water pre-treated were characterized by low microbial counts, following 21 d of storage (by 3.58 and 6.09 log10 CFU g^−1^ lower total plate counts and Pseudomonas spp. counts, respectively, than in control samples)	Segat et al., 2013 [130]
Use of ozone technology in cheese making processes and in storage/maturation condition	The ozone application was conducted at 8 ± 2 °C and for the treatment, an ozone generator with a capacity of 3.5 g ozone h^−1^ was used for 5, 15, 30, and 60 min	Effect of ozone on butter samples	The results showed a higher microbiological quality in butter from raw cream with ozone treatment	Sert et al., 2020 [112]
	Concentrations of 2 and 4 ppm for 10 min were used against *L. monocytogenes* and resident microbiota of cheese rind samples stored at 4 °C for 63 days	Effect of ozone on *Listeria monocytogenes* contamination and the resident microbiota on Gorgonzola cheese rind	In ozonized rinds, the final loads of *L. monocytogenes* were ~1 log CFU g^−1^ higher than controls	Panebianco et al., 2022 [131]
Use of ozone technology in cheese making processes and in storage/maturation condition	For the experiment at pilot plant scale the ripening was conducted in 8 m^3^ ripening rooms for 30 days with gaseous ozone at concentrations of 0, 2, and 6 mg m^−3^. For the experiment at dairy plant scale, a total of 30 cheeses with 15 days of ripening were subjected to maturation with gas ozone at concentrations of 2 mg m^−3^ for 45 days	Effect of gas ozone in soft cheese ripening, in particular its effect on the rind microorganisms	The results showed that gas ozone treatments had a significant fungistatic effect on cheese surface molds for pilot and industrial scale	Tabla and Roa, 2022 [132]
	Pecorino cheese samples were treated for 150 days overnight with gaseous ozone (200 and 300 ppb, 12 °C, and 85% R.H. for 8 h per day)	Effect of gas ozone on the mite pest control and on microbiological growth during Pecorino cheese ripening/storage	Starting from 25 days of storage, 200 ppb of ozone reduced mites and bacteria counts; molds and yeasts started from 75 days of storage. Ozone 300 ppb did not have the same positive impact on some aspects of overall quality.	Grasso et al., 2022 [134]

## 4. Criticality: The Impact of Oxidation and Sensory Alterations

Lipid oxidation is a free radical complex chain reaction in which the formation of free radicals, hydroperoxides, and non-radical degradation products were detected. Lipid oxidation in cheese can be easily observed due to the presence of polyunsaturated fatty acids in their structure at high levels and therefore reaction products that can cause human health can occur [135]. In fact, the elevation of peroxide value (POV) indicates the deterioration of nutritional and sensory qualities of dairy products during storage [136] (Figure 4).

The ozonation of food products can therefore be positively or negatively correlated to some rheological and sensory changes of the product and are strictly dependent on the concentrations used and exposure times, as well as on the food matrix treated.

Regarding the dairy industry, in most cases, no detectable changes from a sensorial point of view were detected after the ozone treatment (concentrations and exposure times) [55]. This conflicting trend underlines that the sensory and nutritional alterations strictly depend on the food matrix, as well as the treatment parameters used. The products of the dairy industry are products with a high content of lipids and proteins and, for this reason, are particularly susceptible to oxidation (Table 7).

Kurtz et al. [137] highlighted a significant sensory alteration in a logical dependent concentration in the powdered milk treated with ozone. Spray-dried skim milk powders produced with an ozone level of 32 ppb received considerably lower sensory scores from a trained panel than those manufactured in air with 2 ppb of ozone. Instead, whole milk powders were shown to have more ozone damage than skim samples, implying that the ozone reacting with the milk fat produced unpleasant odors. This trend was confirmed by Ipsen [138], who also underlined how powdered whole milk was more susceptible to lipid peroxidation than skimmed milk. Uzun et al. [139] studied the changes induced in whey proteins: researchers have highlighted improvements in the foaming power of ozonated whey proteins and in the stability of the foam, due to the greater flexibility of the protein structure, but also a reduction in the solubility and stability of the emulsions. The treatment was conducted with gaseous ozone (60 g h^−1^) or aqueous ozone (4.5 ppm) for up to 15 min.

Segat et al. [140] studied the changes induced in serum proteins by treatments with high concentrations of ozone, noting the same trend highlighted by Uzun et al. [139]: an increase in foaming power and reduction in protein solubility. Furthermore, they highlighted that the treatment causes an increase in the alpha-helix structure and a reduction of the free -SH groups, related to an increase in the surface hydrophobic character. They used a treatment with approximately 20 mg L^−1^ of ozone for 30–480 min.

Sert and Mercan [111] evaluated the effect of ozone on milk and whey proteins, highlighting on the one hand a decrease in the consistency of the product, and on the other an increase in the diameter of the particles, due to the aggregation of the whey proteins milk. The viscosity changed—in the case of milk it increased, while for whey proteins the trend was the opposite. Finally, an increase in brightness and a shift from yellow to blue °hue was recorded in color measurement.

Alexopoulos et al. [129], subjecting the yogurt and Feta samples treated with ozone for 30 and 60 min to sensory analysis by a panel test, verified that, in the case of yogurt, no sensory differences were detected statistically significant, while in the case of Feta the sensory quality was negatively influenced by the 60-min ozone treatment, above all to the detriment of the flavor, but also of the texture and color of the cheese.

Morandi et al. [71] have found, following ozone treatment (4 ppm) of samples of Taleggio DOP and Gorgonzola DOP, a slowdown in the ripening process, linked to the inhibition of the lipolytic process following the ozonation treatment. The sensorial characteristics evaluation of the cheeses showed that the layer just below the rind was altered in its flavor by a lower presence of compounds resulting from lipolysis (free fatty acids and their derivatives), and few oxidation products were found.

Instead, in the study conducted by Grasso et al. [134] on Pecorino cheese, the sensory analysis (consumer test) showed no specific defects with the ozone-treated samples.

An interesting study by Sert et al. [112] highlights the changes induced in butter, a lipid food which is therefore at a high risk of peroxidation, following ozone treatment carried out on the milk cream used for production. Although it has shown itself to be extremely effective in reducing microbial loads, it has been verified that the treatment, depending on the exposure time, causes an increase in the stability, consistency, and size of the fat globules (which, following ozonation, tend to aggregate, presenting a greater tendency to crystallization), with the latter change entailing a reduction in the centrifugation times to separate the fat mass from the aqueous phase. Spreadability, a very important property for the consumers, increases for treatments at 5 and 15 min; moreover, based on the concentration, temperature, and duration of the treatment, this can help to develop the typical butter aroma. The color also undergoes variations: there is an important reduction of the parameter b in the CIELAB color space, which results in a shift of the °hue from yellow to blue, due to a reduction in the carotenoid content induced by ozone. Finally, the oxidation stability in the treated samples decreases with respect to the control sample.

In further research, Sert and Mercan [141], using ozone-treated water in the churning step (0-control and 0.15, 0.20, 0.25, and 0.30 mg L^−1^ ozone samples), observed a decrease in the texture and an increase in the sheen of the butter, a reduction in the diameter of the particles and therefore a reduced spreadability. Churning with ozone treatments increased the redness value of samples and decreased yellowness.

Further studies are needed to identify the ideal concentration–time combinations so that, on the one hand, the antimicrobial effect of ozone can be exploited without incurring sensory defects due to its pro-oxidant action, and, on the other, its ability to modify the characteristics of some molecules to obtain products with specific functions.

## 5. Conclusions

In relation to the dairy field, the potential of using ozone technology is manifold, above all thanks to the versatility of use and environmental compatibility, which is mainly characterized by the absence of release of chemical residues in situ.

The use of ozone technology, in addition to the widely known green role attributed, plays a decisive function, as a sanitizing compound, in the control of microbiological growth, as well of some pathogenic bacteria and other biological pests of a different nature (insects, parasites, viruses).

Nevertheless, more studies need to be performed on some relevant aspects concerning a few still-unknown effects in the cheese production chain. The main topics for further study are:-Determination of doses, exposure times and variables related to the food matrix to be treated in the dairy chain;-Study and characterization of the chemical-physical reactions that take place during the ozonation process;-Focus on the ozone penetration into the desired substrate;-Deepening of the structural/conformational study of the behavior of whey proteins which, following ozonation, see their structure and therefore their functionality modified;-Insight into the sensory, nutritional, and rheological changes that ozonation can cause on milk and its derivatives;-Study of the chemical reactions that can lead to the modification of some molecules (after oxidation with ozone), especially to understand their possible impact on human health;-Improved efficiency of ozone generation systems and study of combined technologies, which associate ozone with other techniques, improving the economic feasibility of treatments.

## Figures and Tables

**Figure 1 foods-12-00987-f001:**
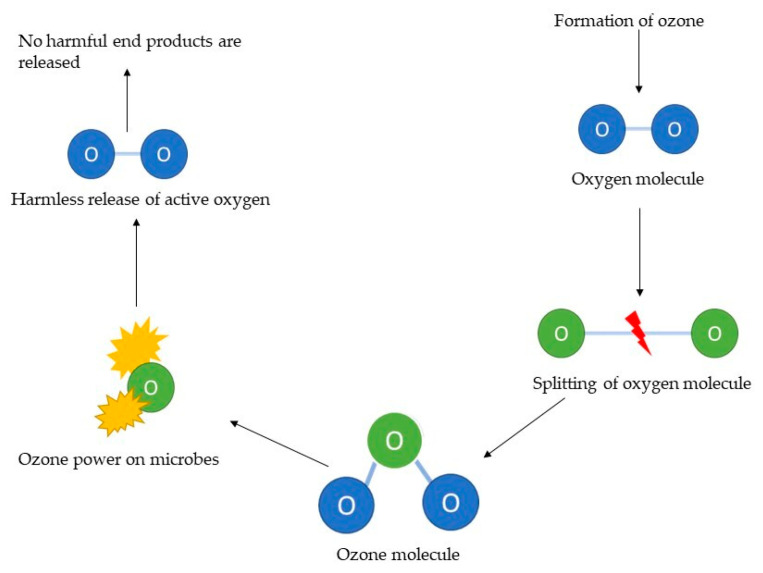
Ozone residues.

**Figure 2 foods-12-00987-f002:**
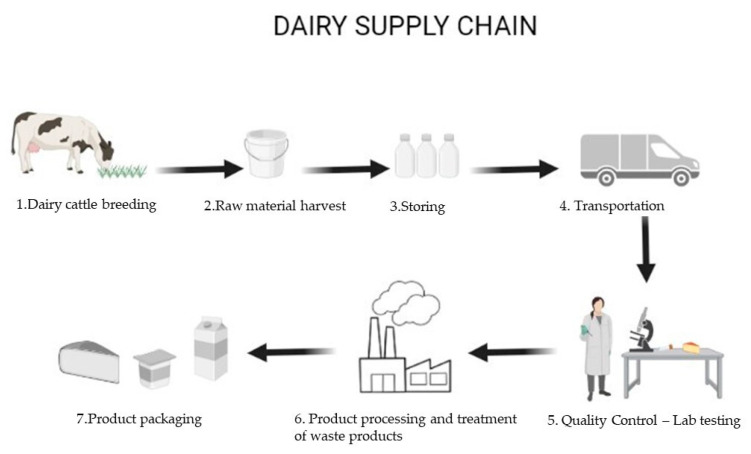
Dairy process flow diagram.

**Figure 3 foods-12-00987-f003:**
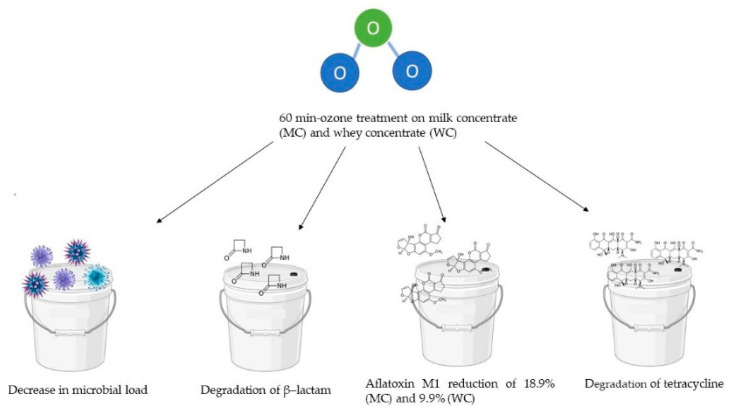
Main results after 60 min. of ozone treatment on milk and whey concentrate.

**Figure 4 foods-12-00987-f004:**
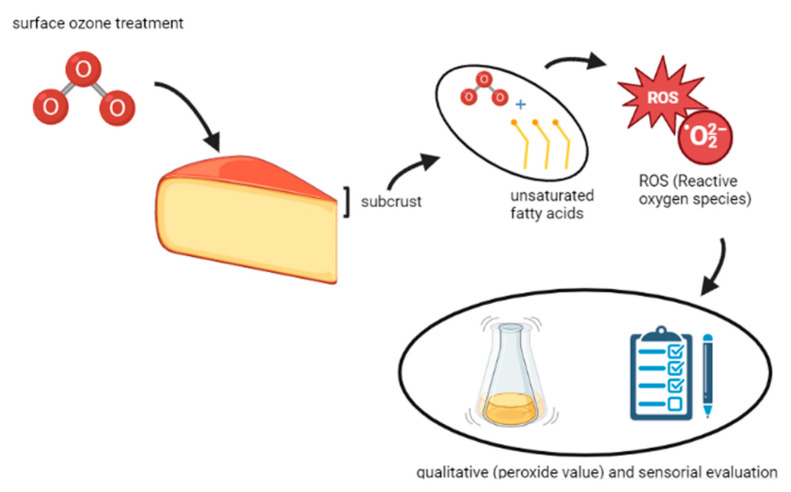
Mechanism of sensory alterations by ozone.

**Table 1 foods-12-00987-t001:** Tons of cheeses produced by the main world countries.

Dairy, Cheese (Tons)			
Country	2019	2020	2021
European Union	10,155	10,232	10,350
United States	5959	6012	6206
Russia	983	1059	1075
Brazil	770	790	790
Argentina	523	488	544
Canada	515	523	540
United Kingdom	472	488	505

**Table 2 foods-12-00987-t002:** The main European cheese-producing nations.

UE-27: Cheese Production (Tons)			
Country	2019	2020	2021
Germany	2,389,288	2,448,640	2,461,334
France	1,695,650	1,671,450	1,716,120
Italy	1,123,640	1,181,390	1,197,390
Total UE-27	9,159,958	9,389,060	9,503,194
% Variation on the previous year	+1.0%	+2.5%	+1.2%

**Table 5 foods-12-00987-t005:** Ozone for wastewater treatment.

Area	Treatments	Target	Result	References
Ozone in wastewater treatment	Ozone concentration of 30 mg dm^−3^ at 25 °C for 5 min	Reduce the content of organic pollutants in wastewater	Thanks to its microflocculation effect, the effectiveness of the following nanofiltration phase increases and, consequently, the reduction of COD is performed; in addition, there is a 40% increase in the biodegradability of nanofiltration residues	Làszlò et al., 2009 [117]
	Ozonation combined with hydrogen peroxide (30% *w*/*w*, [H_2_O_2_] = 9.007 mol L^−1^ and density of 1.1 g mL^−1^) and catalyzed by manganese (1.71 g L^−1^) in alkaline conditions (MnSO_4_.H_2_O, 98% of purity)	Degradation of organic matter in synthetic dairy wastewater	The optimal condition for the ozonation catalyzed by manganese at alkaline medium (C.O.D. removal of 69.4%) can be obtained in pH 10.2 and Mn^2+^ concentration of 1.71 g L^−1^, with COD removals above 60%.	dos Santos Pereira et al., 2018 [118]
Ozone in wastewater treatment	20 min of reaction time with Mn-Fe-Ce/γ-Al_2_O_3_ catalyst, which has a dosage of 12.5 mg·L^−1^·min^−1^ of ozone. pH = 9, and the catalyst dosage was 15 g·L^−1^	Mn-Fe-Ce/γ-Al_2_O_3_ for catalytic ozonation of dairy farm wastewater	The COD removal ratio of dairy farming wastewater can reach 48.9%. The BOD/COD increased from 0.21 to 0.54. Therefore, when the catalyst dosage increases (0–25 g L^−1^), the C.O.D. decreases	Li et al., 2020 [119]
	Five levels of ozonation flow rate (1, 2, 3, 4, 5 L min^−1^) were tested under the conditions that pH was 7.5 and reaction time was 60 min	Effect of ozone on undiluted dairy farm liquid digestate that contains high levels of organic matter, chromaticity and total ammonia nitrogen	After cascade pre-treatment, TAN, TN, COD, and chromaticity were reduced by 80.2%, 75.4%, 20.6%, and 75.8% respectively	Zhu et al., 2022 [120]
	All pre-treatments were performed with ultrasound (US 200 W), ozone (4.2 mg O_3_ L^−1^) and US combined with ozone (US/ozone) for 10 min, 20 min, and 30 min, respectively	Dairy wastewater has been pre-treated with ultrasound, ozone and US combined with ozone to study the fate of enteric indicator bacteria and antibiotic resistance genes, and anaerobic digestion.	US/ozone pre-treatment was effective in the inactivation of enteric indicator bacteria. Total coliforms and enterococci were reduced by 99% and 92% after 30 min	Chen et al., 2021 [121]
Ozone in wastewater treatment	Continuous type O_3_ treatment system (43.26, 87.40, and 132.46 mg L^−1^)	Continuous type O_3_ treatment system to eliminate pathogens such as *Salmonella Typhimurium* and *Escherichia coli* O157: H7 in liquid dairy waste	Ozone reduced *E. coli* O157: H7 and *S. typhimurium* and the reductions increased with the exposure time, particularly at 87.40 or 132.46 mg·L^−1^	Chang et al., 2022 [122]

**Table 7 foods-12-00987-t007:** Oxidation and possible sensory alterations.

Area	Treatments	Target	Result	References
Criticality: oxidation and possible sensory alterations	Ozone concentration of 32 ppb and 2 ppb	Sensory alterations caused by ozone treatment of powdered milk	Spray-dried skim milk powders produced with an ozone level of 32 ppb received considerably lower sensory scores than those manufactured in air with 2 ppb of ozone. Whole milk powders had more ozone damage than skim samples, implying that the ozone reacting with the milk fat produced unpleasant odors	Kurtz et al.,1969 [137]
	Treatment with gaseous ozone with unknown parameters	Effects on powdered whole milk	More susceptible to lipid peroxidation than skimmed milk	Ipsen, 1989 [138]
	The treatment was conducted with gaseous ozone (60 g h^−1^) or aqueous ozone (4.5 ppm) for up to 15 min	Sensory evaluation after ozone treatment on whey protein isolate	Improvements in the foaming power of ozonated whey proteins and in the stability of the foam, due to the greater flexibility of the protein structure, but also a reduction in the solubility and stability of the emulsions	Uzun et al., 2012 [139]
Criticality: oxidation and possible sensory alterations	Treatment with approximately 20 mg L^−1^ of ozone for 30–480 min	Changes induced in serum proteins by treatments with high concentrations of ozone	Increase of the foaming power and reduction of the solubility of the proteins	Segat et al., 2014 [140]
	Ozone treatment was performed to skim milk powder samples at 24 ± 2 °C for 30, 60, 90, and 120 min. Ozone was produced using an ozone generator, which has a capacity of 3.5 g ozone h^−1^	Effect of ozone on milk and whey proteins	Ozone causes a decrease in the consistency of the product, an increase in the diameter of the particles, an increase in the viscosity of the milk, while for whey proteins the trend was the opposite. Finally, in the color measurement there was an increase in brightness and a shift from yellow to blue	Sert and Mercan, 2021 [111]
	Ozone stream (2.5–3 ppm) for 0, 10, 30, and 60 s on the surface of freshly filled yogurt cups before storage for the development of the curd (24 h). Ozone gas was supplied for 0, 10, 20, and 30 min to the brine, where the cheese sample was then left for ripening (2 months)	Sensory evaluation of ozone in yogurt and Feta samples	In the case of Feta, the sensory quality was negatively influenced by the 60-min treatment, above all to the detriment of the flavor, but also of the texture and color of the cheese, while in yogurt no sensory differences were detectable	Alexopoulos et al., 2017 [129]
Criticality: oxidation and possible sensory alterations	Ozone concentration of 4 ppm for 8 min	Investigate the effect of ozone on Ricotta Salata, Taleggio DOP and Gorgonzola DOP cheeses	The layer just below the rind was modified in its flavor by a lower presence of compounds derived from lipolysis (free fatty acids and their derivatives), and fewer oxidation products were found in the same layer	Morandi et al., 2009 [71]
	Pecorino cheese samples were treated for 150 days overnight with gaseous ozone (200 and 300 ppb, 12 °C and 85% R.H. for 8 h per day)	Effect of gas ozone on the mite pest control and on microbiological growth during Pecorino cheese ripening/storage	Sensory analysis (consumer test) showed no specific defects with the ozone-treated samples	Grasso et al., 2022 [134]
	The ozone application was conducted at 8 ± 2 °C and for the treatment, an ozone generator with a capacity of 3.5 g ozone h^−1^ was used for 5, 15, 30, and 60 min	Sensory analysis on butter following ozone treatment	The spreadability and the typical butter aroma are increased. The color undergoes a hue shift from yellow to blue due to an ozone-induced reduction in carotenoid content. Finally, the oxidation stability in the treated samples decreases with respect to the control sample	Sert et al., 2020 [112]
Criticality: oxidation and possible sensory alterations	Butter samples were produced by churning with ozonated waters at concentrations of 0 (control), 0.15 (OW-15), 0.20 (OW-20), 0.25 (OW-25), and 0.30 (OW-30) mg L^−1^. The water temperature during ozonation was at 2 ± 2 °C, and the time of churning was about 30 min for butter production	Effects of ozonated water on particle size, texture, oxidation, melting and microbiological characteristics of butter	A decrease in texture and an increase in the sheen of the butter, a reduction in the diameter of the particles, and, therefore, a reduced spreadability	Sert and Mercan, 2020 [141]

## Data Availability

Data is contained within the article.

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
