# Peer review of "The Use of Ozone Technology: An Eco–Friendly Method for the Sanitization of the Dairy Supply Chain"

_foods, 2023, doi:10.3390/foods12050987_

Round 1
Reviewer 1 Report
The Use of Ozone Technology: an Eco – Friendly Method for the Sanitization of the Dairy Supply Chain
This is an outline review of interest but require a significant further improvement. Not sufficiently illustrative figure also not critical appraisal of literature & number of grammatical errors. There are several out of date references in this manuscript. The authors should consider updating them with the newest developments. There were several new studies on ozone's application in the dairy industry that emerged, particularly after the pandemic
The authors should consult more recent reviews that give description of ozone's properties, and inactivation mechanism. The conclusion can be further improved also future directions as result of this review would enhance overall impact of review.
Grammatical error: errors of this kind persist throughout the document.
17 Line; “like the treatment of wastewater. Ozone is ease for use and eco‐sustainable as it”
What is this “alterative effect”
19 “to an alterative effect”
What chemicals
58 “more chemicals for cleaning compared to”
What is this amount & ref
60-65 “The huge amounts of wastewater”
It may be important to mention safety concerns regarding the use of Chlorine in food processing. Please check the restrictions of Chlorine usage in the EU and the UK
64- 65; Sanitation of water with chlorine gas (Cl2), dioxide (ClO2), or hypochlorite (ClO-) remains common practice in many jurisdictions due to chlorine's bactericidal and oxidative properties.
Would also suggest to put in information regarding what main microbiota are present & in what concentration as a table etc Also how “useful” microorganism can be preserved etc
77- 85; “to contamination by spoilage microorganisms”
It is important to create this balance. Some disinfection by-products which are toxic may be formed. The authors should briefly discuss (with an illustration) how this known issue with ozonation affects the dairy industry.
98- 100; “Unlike other disinfectants, it leaves no chemical residuals and
degrades to molecular oxygen upon reaction or natural degradation”
This figure is not illustrative enough, can be further improved, also not sufficient information in text
Figure 1. Dairy supply chain.
No reference is given here for ozone's properties. The authors should consult more recent reviews in the field (e.g. Ozone application in different industries: A review of recent developments, CEJ, 454, 140188), that give a detailed description of ozone's properties, and inactivation mechanism
119: Ozone has a molecular weight”
Require a subscript
143; (mg m‐3)
Into atomic what??
145; “into atomic”
The use of “attitude” is incorrect. Behaviour is a more appropriate term
153: an electrophilic attitude the other
Unnecessary full stop etc
200; various studies.
An important industrial factor not captured here is that of ozone penetration into the desired substrate. The following studies can be used to highlight this crucial point
https://doi.org/10.1080/01919512.2022.2066503
235 – 2337 ; Lastly, very significant is the use in fruit and vegetable production; in this context
Recently (https://doi.org/10.1016/j.mimet.2022.106431), ozone has also been shown to be effective against Aspergillus fumigatus (a very difficult fungus to inactivate using other methods). It is worth including this point
295; “In summary, it turns out that”
What does this “ aw “ mean
315; influencing the aw and therefore the possibility of
Sentence to be modified as it require to explain why copper & steel
383; Since metal surfaces, the corrosive power of ozone must always be considered, for the occurrence of copper and carbon steel materials.
Recommend further information on the effect on materials of ref https://doi.org/10.1021/acsomega.2c05264
384; “ozonated water can only be recommended to replace warm water and chlorine for”
As ozone is non-selective towards harmful or useful bacteria. It becomes a challenging when Lactic acid bacteria (LAB) - one of the most significant groups of probiotic organisms, commonly used in fermented dairy products is destroyed by ozone's application. The authors should comment on this point, as it related to the preservation of dairy products by ozone.
3.4 Ozone treatment in milk production
The use of a figure might be better to explain these results.
528 – 540; “Volume mean diameter of”
Update references in Tables 3 and 4 to include most recent
Table 4. Ozone milk treatment
As this is a crucial point. Consider including a figure in this section illustrating the mechanism of sensory alterations or a collage of key results of 2-3 studies. This will draw the attention of reader to this important point.
4. Criticality: the impact of oxidation and sensory alterations
This statement seems misplaced. I don't think it is meant to be in the conclusion section.
857- 858; The cheese is considered a multifactorial biological system consisting of a hetero-genous classes of compounds (fats, proteins and carbohydrates) in a complicated physical matrix.
Author Response
1 Comment:17 Line “like the treatment of wastewater. Ozone is ease for use and eco‐sustainable as it”. Grammatical error.
Answer: Done in the text. Line 18.
2 Comment :19 Line “to an alterative effect”. What is this “alterative effect”?
Answer: Changed in “However, its oxidation potential can lead to the peroxidation of cheese polyunsaturated fatty acids.”. Lines 19-20.
3 Comment: 58 Line “more chemicals for cleaning compared to”. What chemicals?
Answer: Done in the text. Lines 56-58.
4 Comment: Line 60-65 “The huge amounts of wastewater”. What is this amount & ref?
Answer: Thank you very much for this observation. Added in the text (lines 61-70).
5 Comment: Line 64- 65; Sanitation of water with chlorine gas (Cl2), dioxide (ClO2), or hypochlorite (ClO-) remains common practice in many jurisdictions due to chlorine's bactericidal and oxidative properties. It may be important to mention safety concerns regarding the use of Chlorine in food processing. Please check the restrictions of Chlorine usage in the EU and the UK.
Answer: Done in the text. Lines 70-83.
6 Comment: Lines 77- 85; “to contamination by spoilage microorganisms”. Would also suggest to put in information regarding what main microbiota are present & in what concentration as a table etc Also how “useful” microorganism can be preserved etc
Answer: Done in the text. Lines 101-118.
7 Comment: Lines 98- 100; “Unlike other disinfectants, it leaves no chemical residuals and degrades to molecular oxygen upon reaction or natural degradation”. It is important to create this balance. Some disinfection by-products which are toxic may be formed. The authors should briefly discuss (with an illustration) how this known issue with ozonation affects the dairy industry.
Answer: Thank you very much for this observation. Done in the text. Lines 133-135.
Image added in the text (line 138).
8 Comment: Figure 1. Dairy supply chain. This figure is not illustrative enough, can be further improved, also not sufficient information in text.
Answer: Image added on the text (line 153).
9 Comment: Line 119: Ozone has a molecular weight”. No reference is given here for ozone's properties. The authors should consult more recent reviews in the field (e.g. Ozone application in different industries: A review of recent developments, CEJ, 454, 140188), that give a detailed description of ozone's properties, and inactivation mechanism
Answer: Thank you for this observation. Text added. Lines 165-168.
10 Comment: Line 143; (mg m‐3). Require a subscript.
Answer: Done in the text. Line 187.
11 Comment: Line 145; “into atomic”. Into atomic what?
Answer: Done on the text. Line 190.
12 Comment: Line 153: an electrophilic attitude the other. The use of “attitude” is incorrect. Behaviour is a more appropriate term.
Answer: Done in the text. Line 197.
13 Comment: Line 200; various studies. Unnecessary full stop etc.
Answer: Thank you. Deleted in the text.
14 Comment: Lines 235 – 237 ; Lastly, very significant is the use in fruit and vegetable production; in this context. An important industrial factor not captured here is that of ozone penetration into the desired substrate. The following studies can be used to highlight this crucial pointhttps://doi.org/10.1080/01919512.2022.2066503
Answer: Done in the text. Lines 294-309.
15 Comment: Line 295; “In summary, it turns out that”. Recently (https://doi.org/10.1016/j.mimet.2022.106431), ozone has also been shown to be effective against Aspergillus fumigatus (a very difficult fungus to inactivate using other methods). It is worth including this point.
Answer: Thank you very much for this observation. Done in the text. Lines 363-366.
16 Comment: Line 315; influencing the aw and therefore the possibility of. What does this “aw “ mean.
Answer: Done in the text. Line 385.
17 Comment: Line 383; Since metal surfaces, the corrosive power of ozone must always be considered, for the occurrence of copper and carbon steel materials. Sentence to be modified as it require to explain why copper & steel.
Answer: Done in the text. Lines 439-445.
18 Comment: Line 384; “ozonated water can only be recommended to replace warm water and chlorine for”. Recommend further information on the effect on materials of ref https://doi.org/10.1021/acsomega.2c05264
Answer: Done in the text. Lines 445-452.
19 Comment: Paragraph 3.4 Ozone treatment in milk production. As ozone is non-selective towards harmful or useful bacteria. It becomes a challenging when Lactic acid bacteria (LAB) - one of the most significant groups of probiotic organisms, commonly used in fermented dairy products is destroyed by ozone's application. The authors should comment on this point, as it related to the preservation of dairy products by ozone.
Answer: Done in the text. Lines 734-743.
20 Comment: Lines 528 – 540; “Volume mean diameter of”. The use of a figure might be better to explain these results.
Answer: Thank you very much for this observation. We believe that the image can improve our article. Image added on the text (line 600).
21 Comment: Table 4. Ozone milk treatment. Update references in Tables 3 and 4 to include the most recent.
Answer: Thank you very much for this observation. We have already selected the studies most relevant over the years and all the works of recent years.
22 Comment: Paragraph 4. Criticality: the impact of oxidation and sensory alterations. As this is a crucial point. Consider including a figure in this section illustrating the mechanism of sensory alterations or a collage of key results of 2-3 studies. This will draw the attention of reader to this important point.
Answer: Thank you very much for this observation. We believe that the image can improve our article. Image added on the text (line 834).
23 Comment: Lines 857- 858; The cheese is considered a multifactorial biological system consisting of a hetero-genous classes of compounds (fats, proteins and carbohydrates) in a complicated physical matrix. This statement seems misplaced. I don't think it is meant to be in the conclusion section.
Answer: Deleted.

Reviewer 2 Report
In my opinion the article needs some improvements.
Some examples of improvements are presented below:
- The abstract should not have more than 200 words.
- Row 64 - Please add Chlorine before dioxide (ClO2).
- Row 70 - Please explain the acronym CIP – Cleaning In Place.
- Row 77 - Please remove ; [11;12], and add,
- Row 90 - Please remove ; [14;15;16], and add,
- Row 169 - Please explain the acronyms: PTFE, PVDF, PVC, ECTFE – Polytetrafluoroethylene, Polyvinylidene fluoride, Polyvinyl chloride, ethylene chlorotrifluoroethylene.
- Row 239 - Please remove ; [30;31;32;33], and add, / Please look for this aspect in the whole article.
- Row 250 - Please explain the acronyms.
- Row 281 - Gram‐positive - with a capital G
- Please write the names of the microorganisms either using the full name (e.g. Listeria monocytogenes) or the short one (e.g. L. monocytogenes). / Please look for this aspect in the whole article.
- Row 297 - Gram‐negative - with a capital G
- Row 315 - Please explain aw - water activity index.
- Row 378 - Greene et al., [57] - Please remove ,
- Row 474 - Jorek et al., [72] - Please remove , / Please look for this aspect in the whole article.
- CFU or cfu? / Please look for this aspect in the whole article.
- Row 562 Mn2+ (2+ at superscript) / Please look for this aspect in the whole article.
- Row 624 - Please explain the acronym when it appears for the first time in the text. / Please look for this aspect in the whole article.
- References should be prepared in accordance with the requirements of the journal.
Author Response
1 Comment: The abstract should not have more than 200 words.
Answer: The abstract has 200 words.
2 Comment: Row 64 - Please add Chlorine before dioxide (ClO2).
Answer: Thank you very much for this observation. Added in the text (line 71).
3 Comment: Row 70 - Please explain the acronym CIP – Cleaning In Place.
Answer: Added in the text (line 88).
4 Comment: Row 77 - Please remove ; [11;12], and add,
Answer: Thank you. Removed and added in the text (line 95).
5 Comment: Row 90 - Please remove ; [14;15;16], and add,
Answer: Checked in the whole article.
6 Comment: Row 169 - Please explain the acronyms: PTFE, PVDF, PVC, ECTFE – Polytetrafluoroethylene, Polyvinylidene fluoride, Polyvinyl chloride, Ethylene Chlorotrifluoroethylene.
Answer: We explained the acronyms in the text (lines 212-213).
7 Comment: Row 239 - Please remove ; [30;31;32;33], and add, / Please look for this aspect in the whole article.
Answer: Checked in the whole article.
8 Comment: Row 250 - Please explain the acronyms.
Answer: Explained in the text (line 319-320).
9 Comment: Row 281 - Gram‐positive - with a capital G.
Answer: Thank you very much for this observation. Modified in the whole text.
10 Comment:. Please write the names of the microorganisms either using the full name (e.g. Listeria monocytogenes) or the short one (e.g. L. monocytogenes). / Please look for this aspect in the whole article
Answer: Done in the whole text.
11 Comment: Row 297 - Gram‐negative - with a capital G.
Answer: Modified in the text.
12 Comment: Row 315 - Please explain aw - water activity index.
Answer: Done in the text (line 385).
13 Comment: Row 378 - Greene et al., [57] - Please remove ,
Answer: Done in the text (line 439).
14 Comment: Row 474 - Jorek et al., [72] - Please remove , / Please look for this aspect in the whole article.
Answer: Thank you very much for this observation. Done in the whole article.
15 Comment: CFU or cfu? / Please look for this aspect in the whole article.
Answer: CFU, done in the whole text.
16 Comment: Row 562 Mn2+ (2+ at superscript) / Please look for this aspect in the whole article.
Answer: Done in the text (line 623) and checked in the whole text.
17 Comment: Row 624 - Please explain the acronym when it appears for the first time in the text. / Please look for this aspect in the whole article.
Answer: Thank you very much for this observation. PDO (Protected Designation of Origin) is explained on lines 683-684. CFU (colony-forming units) is explained in line 501.
18 Comment: References should be prepared in accordance with the requirements of the journal.
Answer: Done. See the text.

Reviewer 3 Report
1. Please consider the staff exposure and health effects of ozone application.
2. Please write the methodology in the abstract.
3. Delete the excess keywords.
4. Please rewrite the introduction using related sentences including dairy disinfection and sanitation.
5. Please consider the byproduct of ozonation.
6. Ecological footprint of ozone should be considered.
7. Define the abbreviations such as: CIP
8. Please provide the process flow diagram of dairy industry.
9. Please provide the reference in P:5,line:138-146
10. P:5, line:148, Phrase “reactive oxygen species (ROS or 148 Reactive Oxygen Species),”has been repeated.
11. You can use the following article in scope of disinfection in your paper:
· M. A. Mirnasab, Hassan Hashemi, M. R. Samaei, A. Azhdarpoor. Advanced removal of water NOMs by pre-ozonation, Enhanced coagulation and bio-augmentated granular activated carbon. International journal of environmental science and technology, 01 January 2021
Author Response
1 Comment: Please consider the staff exposure and health effects of ozone application.
Answer: Thank you very much for this observation. We believe it can improve our article. Text added (lines 221-242).
2 Comment: Please write the methodology in the abstract.
Answer: Done in the text (lines 20-21).
3 Comment: Delete the excess keywords.
Answer: Deleted “green technologies” (line 22).
4 Comment: Please rewrite the introduction using related sentences including dairy disinfection and sanitation.
Answer: Thank you very much for this observation. We believe it can improve our article. The introduction has been revised.
5 Comment: Please consider the byproduct of ozonation.
Answer: Added on the text (lines 133-135).
6 Comment: Ecological footprint of ozone should be considered.
Answer: Thank you, but no works related to the ozone footprint were found. An image on the formation of ozone, reaction and release of oxygen into the environment has been included in the review (line 138).
7 Comment: Define the abbreviations such as: CIP.
Answer: Done in the text (line 88).
8 Comment: Please provide the process flow diagram of dairy industry.
Answer: Thank you very much for this observation. Done. Fig. 2 (line 153).
9 Comment: Please provide the reference in P:5,line:138-146
Answer: Thank you very much for this observation. Reference added in the text (line 190).
10 Comment: P:5, line:148, Phrase “reactive oxygen species (ROS or 148 Reactive Oxygen Species),”has been repeated.
Answer: Done in the text (lines 192-193).
11 Comment: You can use the following article in scope of disinfection in your paper:
- M. A. Mirnasab, Hassan Hashemi, M. R. Samaei, A. Azhdarpoor.Advanced removal of water NOMs by pre-ozonation, Enhanced coagulation and bio-augmentated granular activated carbon. International journal of environmental science and technology, 01 January 2021
Answer: Thank you. Done in the text (lines 272-279).

Round 2
Reviewer 3 Report
Corrections are acceptable.